# Provable Ordering and Continuity in Vision-Language Pretraining for Generalizable Embodied Agents

**Zhizhen Zhang**
The University of Queensland
uqzzha39@uq.edu.au

**Lei Zhu**
Tongji University
leizhu0608@gmail.com

**Zhen Fang**
University of Technology Sydney
zhen.fang@uts.edu.au

**Zi Huang**
The University of Queensland
helen.huang@uq.edu.au

**Yadan Luo**
The University of Queensland
y.luo@uq.edu.au

## Abstract

Pre-training vision-language representations on human action videos has emerged as a promising approach to reduce reliance on large-scale expert demonstrations for training embodied agents. However, prior methods often employ time contrastive learning based on goal-reaching heuristics, progressively aligning language instructions from the initial to the final frame. This overemphasis on future frames can result in *erroneous* vision-language associations, as actions may terminate early or include irrelevant moments in the end. To address this issue, we propose Action Temporal Coherence Learning (AcTOL) to learn *ordered* and *continuous* vision-language representations without rigid goal-based constraint. AcTOL treats a video as a continuous trajectory where it (1) contrasts semantic differences between frames to reflect their natural ordering, and (2) imposes a local Brownian bridge constraint to ensure smooth transitions across intermediate frames. Extensive imitation learning experiments on both simulated and real robots show that the pretrained features significantly enhance downstream manipulation tasks with high robustness to different linguistic styles of instructions, offering a viable pathway toward generalized embodied agents. Our project page is at https://actol-pretrain.github.io/.

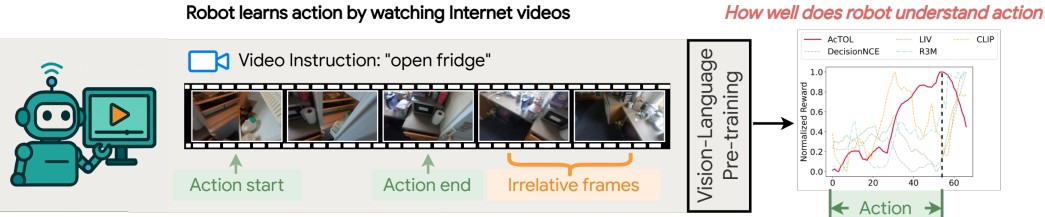

Figure 1: Pretraining on Internet human action videos for robot control, where the video-instruction pairs are noisy and often include irrelevant frames. The red vision-language reward curve demonstrates AcTOL learns to correctly align instruction with action, outperforming previous goal-reaching methods in the presence of distracting content.

## 1   Introduction

The long-term vision for embodied intelligence [29, 25] is to create systems that seamlessly perceive and interact with the world around them. Achieving this requires agents that integrate vision and language to understand their surroundings, interpret human instructions, and autonomously plan

39th Conference on Neural Information Processing Systems (NeurIPS 2025).

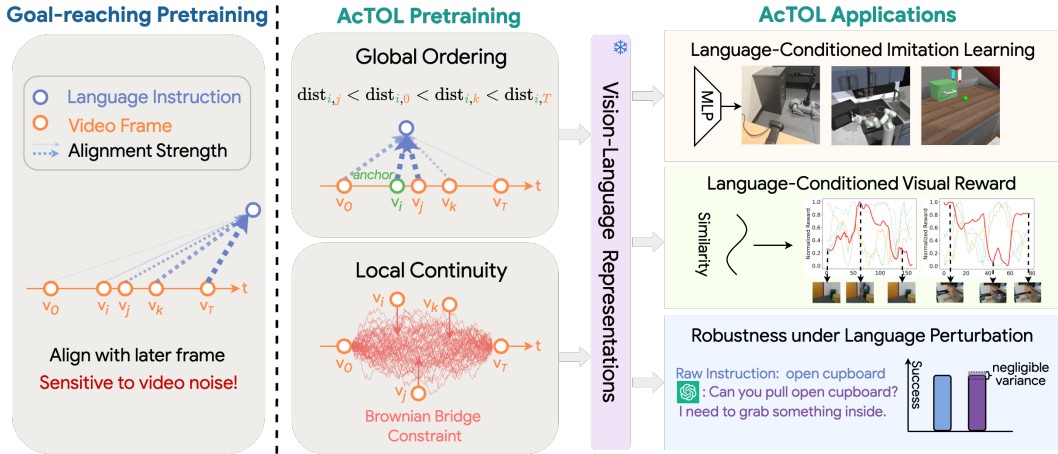

Figure 2: Comparison of existing *goal-reaching* pre-training strategies and the proposed AcTOL approach. Our learned multi-modal representations can be effectively transferred to downstream language-conditioned robot manipulation tasks, exhibiting robustness to diverse instruction and linguistic variations.

actions for complex tasks. Current end-to-end approaches achieve policy learning through direct vision-language-action mapping [48, 13, 7, 21, 3]. However, the inherent unpredictability of physical environments, including unseen scenarios and dynamic object interactions, constrains these solutions by requiring massive, high-quality robotic trajectories with action annotations, which are *costly* to collect. To mitigate this, recent research has leveraged large-scale, readily available egocentric human action videos [14, 10, 15] for *pre-training*. Although these out-of-domain videos often lack low-level action details and contain noise, their diverse human-object interactions and task instructions provide valuable prior knowledge. This enables the pre-trained representations to be more effectively transferred to novel tasks with fewer demonstrations, reducing reliance on large-scale robotic datasets while preserving strong generalization capabilities.

A promising approach for vision-language pre-training from human action videos leverages the concept of *time contrastive learning* [37] to capture temporally consistent visual representations, where language serves as the guiding goal, with semantic alignment between the language and chronologically later frames in the video [30, 26, 23]. However, this *goal-reaching* semantic alignment approach relies on a rigid assumption that action videos adhere to a specific principle: *actions progressively approach the target instruction from the initial frame to the final one*. Such assumption can be easily violated in egocentric human action videos, which are typically annotated at a coarse-grained level and riddled with noise. Figure 1 shows an example video-instruction pair, where the end of the video clip does not correspond to the actual end of the action. As a result, existing methods suffer from misleading semantic alignment, which hampers their ability to learn accurate vision-language relationships.

Given the challenges outlined above, a more natural and flexible pre-training strategy without rigid assumptions is needed to enhance vision-language representations for better policy learning. Building solely on the intrinsic temporal consistency of human action videos, we argue that the *ordering* and *continuity* of pre-trained vision-language representations play a crucial role in ensuring the effectiveness of policy learning. Ordering refers to the need for visual features to align with the underlying action logic required by the language instruction. For instance, as the task progresses, visual representations closer to the completion of the action should exhibit stronger alignment with the language instruction. This ensures that each step in the sequence is meaningfully associated with the corresponding instruction, enabling the model to effectively capture the dynamic progression of the task. Continuity, on the other hand, emphasizes that both visual features and their alignment with the language should evolve smoothly over time, with gradual transitions rather than abrupt changes. This is crucial because actions in the real world are not discrete but unfold continuously in time. Moreover, the alignment between visual and instruction should also be fluid, ensuring that as the action progresses, the visual representations consistently align with the target language instruction.

To address the aforementioned issues, as illustrated in Figure 2, we propose Action Temporal Coherence Learning (AcTOL), a novel approach designed to implicitly capture the ordering and continuity of video actions without relying on rigid assumptions, while providing strong theoretical

guarantees. Unlike previous approaches that focus on goal-directed semantic alignment, AcTOL introduces a Vision-Language Ordering (VLO) loss. This loss leverages the intrinsic temporal coherence of videos, contrasting frames against each other based on their relative temporal distance, theoretically ensuring that the semantic alignment between frames reflects their temporal ordering and continuity throughout the entire sequence. However, the VLO loss does not explicitly enforce the continuity of the visual features themselves, and under conditions with variations in frame content and noise, it can lead to suboptimal local consistency of the visual features. To address this, AcTOL introduces a Brownian bridge constraint over the video, treating video frames as a Brownian bridge process. This approach imposes a structured, continuous flow on the visual representations, ensuring that the model learns more consistent and stable intermediate states, further enhancing the continuity of the visual representations and improving the stability of their alignment with language instruction. Further theoretical analysis suggests that these properties also contribute to the model's resilience to language perturbations, a crucial trait for real-world applications. To evaluate the generalization ability of AcTOL on embodied agents, we conducted extensive language-conditioned imitation learning experiments using both the real-world Unitree D1 robotic arm and two simulation environments. The results demonstrate that AcTOL significantly outperforms prior methods with a limited number of expert demonstrations. Additionally, AcTOL can generate language-conditioned visual rewards from real-world robot videos and remains robust to complex linguistic perturbations, highlighting its potential as a generalizable solution for real-world embodied agents.

## 2    Preliminaries

We first set up notations and mathematically formulate tasks.

**Language-Conditioned Imitation Learning (LC-IL).** The task of LC-IL aims to train an agent to mimic expert behaviors from a given robot demonstration set $\mathcal{D}_{\text{robot}} = \{(\tau_i, l_i)\}_{i=1}^{N_r}$, where $l_i \in \mathcal{L}$ represents a task-specific language instruction. Each trajectory $\tau_i \in \mathcal{T}$ consists of a sequence of state-action pairs $\tau_i = \{(\mathbf{s}_t, \mathbf{a}_t)\}_{t=1}^T$ of the horizon length $T$. In robot manipulation tasks, action $\mathbf{a}_t \in \mathcal{A}$ corresponds to the control commands executed by the agent and state $\mathbf{s}_t = [\mathbf{p}_t; \mathbf{o}_t] \in \mathcal{S}$ records proprioceptive data $\mathbf{p}_t$ (*e.g.,* joint positions, velocities) and visual inputs $\mathbf{o}_t$ (*e.g.,* camera images) at the time step $t$. The objective of LC-IL is to find an optimal language-conditioned policy $\pi^*(\mathbf{a}|\mathbf{s}, l) : \mathcal{S} \times \mathcal{L} \mapsto \mathcal{A}$ via solving the supervised optimization as follows,

$$\pi^* \in \arg\min_\pi \mathbb{E}_{(\tau_i, l_i) \sim \mathcal{T}} \left[ \frac{1}{T} \sum_{(\mathbf{s}_t, \mathbf{a}_t) \sim \tau_i} \ell(\pi(\hat{\mathbf{a}}_t, \mathbf{s}_t | l_i), \mathbf{a}_t) \right],$$

where $\ell(\cdot, \cdot)$ is a task-specific loss, such as mean squared error or cross-entropy. Training the policy $\pi_\theta$ in an end-to-end fashion may require *hundreds* of high-quality expert demonstrations to converge, primarily due to the high variance of visual inputs $\mathbf{o}$ and language instructions $l$.

**Vision-language Pre-training.**    Address such scalability issues can be achieved by leveraging large-scale, easily accessible human action video datasets [10, 15] $\mathcal{D}_{\text{human}} = \{(\mathcal{O}_i, l_i)\}_{i=1}^{N_h}$, where $\mathcal{O}_i = \{o_t\}_{t=1}^T$ represents a video clip with $T$ frames and $l_i$ the corresponding description. Pretraining on such datasets enables policies to rapidly learn visual-language correspondences with minimal expert demonstrations. Mainstream pretraining methods employ *time contrastive learning* [37] to fine-tune a visual encoder $\phi$ and a text encoder $\varphi$, which project frames and descriptions into a shared $d$-dimensional embedding space, *i.e.,* $\mathbf{v}_t = \phi(o_t) \in \mathbb{R}^d$ and $\mathbf{l}_i = \varphi(l_i) \in \mathbb{R}^d$. To provide a unified perspective on various pretraining approaches, we formulate them within the objective $\mathcal{L}_{\text{tNCE}}(\phi, \varphi)$:

$$\mathcal{L}_{\text{tNCE}} = -\mathbb{E}_{o^+ \sim \mathcal{P}(\mathcal{O}_i)} \log \frac{\exp(\mathfrak{R}(\mathbf{v}^+, \mathbf{l}_i))}{\mathbb{E}_{o^- \sim \mathcal{N}(\mathcal{O}_i)} \exp(\mathfrak{R}(\mathbf{v}^-, \mathbf{l}_i))},$$

where $\mathbf{v}^{+/-} = \phi(o^{+/-})$. Different pretraining strategies differ in their selection of (1) the positive frame set $\mathcal{P}(\mathcal{O}_i)$, (2) negative frame set $\mathcal{N}(\mathcal{O}_i)$; and (3) the semantic alignment scoring function $\mathfrak{R}(\mathbf{v}, \mathbf{l}_i)$ measuring the gap of VL similarities.

As motivated by goal-conditioned RL [1], current approaches *explicitly* select future frames (*e.g.,* R3M [30], DecisionNCE [23]) or the last frame (*e.g.,* LIV [26]) as the goal within the positive frame

set, enforcing their visual embedding to align with the semantics. Likewise, the scoring functions $\mathfrak{R}$ are often designed to maximize this transition direction. However, the pretraining action videos are *noisy* as actions may terminate early or include irrelevant subsequent actions, which may mislead the encoders and result in inaccurate vision-language association. As detecting precise action boundaries is non-trivial, we argue for a more flexible approach that leverages *intrinsic* characteristics of actions to guide pertaining.

## 3  Our Approach: AcTOL

We introduce an action temporal coherence learning (AcTOL) to capture two temporal properties of video actions: *ordering* and *continuity*. *Ordering* was ensured in the vision-language ordering loss (Section 3.1), where the semantic difference between frames reflects their temporal distance, with closer frames exhibiting smaller differences than those further apart. *Continuity* requires smooth visual transitions between adjacent frames, avoiding abrupt changes and high variance. To achieve this, we model sampled frame intervals as a Brownian bridge process (Section 3.2), penalizing deviations from the expected trajectories. Different from prior works that relies on setting explicit goal frames, the proposed approach *implicitly* explore the global and local structure of actions without imposing rigid constraints.

### 3.1  Visual-Language Ordering

To capture the temporal coherence of video actions, we first propose a vision-language ordering (VLO) loss that ensures the semantic alignment between frames reflects their temporal order. Since the VLO loss is applied within each video individually, we henceforth write $\mathcal{O}_i, l_i$ as $\mathcal{O}, l$ for simplicity. Consider an anchor frame $o_i \in \mathcal{O}$ with an index $n(i)$ corresponding to its position in the original video. For any given frame pair $(o_i, o_j)$, we first define the semantic alignment score $\mathfrak{R}$ to quantify differences in their VL similarities *w.r.t* a language description $l$ as:

$$\mathfrak{R}(\mathbf{v}_i, \mathbf{v}_j, \mathbf{l}) = -\| \operatorname{sim}(\mathbf{v}_i, \mathbf{l}) - \operatorname{sim}(\mathbf{v}_j, \mathbf{l}) \|_2, \tag{1}$$

where $\mathbf{v}_i = \phi(o_i), \mathbf{l} = \varphi(l)$. The function $\operatorname{sim}(\cdot, \cdot)$ computes the VL similarity using cosine similarity. To ensure the proposed $\mathfrak{R}$ adhere to the temporal ordering of frames, we construct a negative set $\mathcal{N}_{i,j}$ by selecting $o_k \in \mathcal{O}$ correspond to frames that are temporally more *distant* from $o_i$ than $o_j$:

$$\mathcal{N}_{i,j} = \{o_k \mid k \neq i, \ |n(i) - n(k)| \geq |n(i) - n(j)|\},$$

This formulation allows us to reformulate $\mathcal{L}_{\text{tNCE}}$ by enforcing that the VL similarity difference between frames $i$ and $j$ should be smaller than that between frame $i$ and any negative frame $k$ within the video $\mathcal{O}$:

$$\mathcal{L}_{\text{VLO}} = -\mathbb{E}_{(o_i, o_j) \sim \mathcal{O}} \log \frac{\exp\left(\mathfrak{R}(\mathbf{v}_i, \mathbf{v}_j, \mathbf{l})\right)}{\sum_{o_k \in \mathcal{N}_{i,j}} \exp\left(\mathfrak{R}(\mathbf{v}_i, \mathbf{v}_k, \mathbf{l})\right)}.$$

Notably, our VLO loss does not strictly require $o_j$ to be from a future timestep for goal-reaching. Instead, we leverage the inherent temporal dynamics in videos, allowing the model to learn the natural ordering in an unsupervised manner.

### 3.2  Vision-Language Continuity

While the VLO property provides a strong global constraint on the structural alignment of VL pretraining, optimizing triplet relationships alone can be *unstable*. Variations in frame content and noise often lead to *suboptimal* local consistency. To mitigate this, we introduce an additional local continuity constraint inspired by the *Brownian bridge* [36]. This stochastic process models transitions between two fixed endpoints over by any sampled local video interval $[n(i), n(j)]$. For any time step $t \in [n(i), n(j)]$ within this interval, the transition density of Brownian Bridge process $\mathbf{B}(t)$ follows a time-dependent Gaussian distribution:

$$\mathcal{N}\left(\mathbf{v}_i + \frac{t - n(i)}{n(j) - n(i)}(\mathbf{v}_j - \mathbf{v}_i), \frac{t(n(j) - n(i)) - t^2}{n(j) - n(i)}\right),$$

where $\mathbf{v}_i, \mathbf{v}_j \in \mathbb{R}^d$ are the visual embeddings of the first and last frames in the sampled interval. The mean trajectory $\mathbb{E}[\mathbf{B}(t)]$ linearly interpolates between the two endpoints, while the variance

$\text{Var}[\mathbf{B}(t)]$ provides uncertainty modeling that peaks in the middle of the interval. To enforce this local continuity, the Brownian bridge loss $\mathcal{L}_{\text{BB}}$ is formulated as,

$$\mathcal{L}_{\text{BB}} = \frac{1}{T} \sum_{t=1}^{T} \frac{1}{2\text{Var}[\mathbf{B}(t)]} \left\| \mathbf{v}_t - \mathbb{E}[\mathbf{B}(t)] \right\|_2^2. \tag{2}$$

This loss encourages local consistency by penalizing deviations from expected trajectories, ensuring consistency across short temporal spans.

**Overall Objective.** The final training objective integrates both global and local constraints to achieve temporal coherence simultaneously:

$$\mathcal{L}_{\text{AcTOL}} = \mathcal{L}_{\text{VLO}} + \lambda \mathcal{L}_{\text{BB}}, \tag{3}$$

where $\lambda$ is empirically set to balance two components.

## 4 Theoretical Analysis

In this section, we theoretically prove the vision-language ordering and continuity, as well as extend the robustness of linguistic perturbations of representations learned by AcTOL. All proofs are provided in Appendix C for reference.

**Vision-Language Ordering.** Ordering and sorting properties are well-established in self-supervised learning [39, 18, 45]. Building upon these insights, we formalize the concept of vision-language ordering (VLO) below.

**Definition 1** (VLO Representations). Let $\{o_i\}_{i \in [T]}$ be a sequence of video frames and $l$ the corresponding language description. The representations of the frames are said to satisfy the VLO property for any $0 < \delta < 1$ if $\forall i \in [T]$, and distinct frames $j, k \in [T] \backslash \{i\}$, the following conditions hold:

$$\begin{cases} \mathfrak{R}_{i,j,l} > \mathfrak{R}_{i,k,l} + 1/\delta, & \text{if } d_{i,j} < d_{i,k}, \\ |\mathfrak{R}_{i,j,l} - \mathfrak{R}_{i,k,l}| < \delta, & \text{if } d_{i,j} = d_{i,k}, \\ \mathfrak{R}_{i,j,l} < \mathfrak{R}_{i,k,l} - 1/\delta, & \text{if } d_{i,j} > d_{i,k}, \end{cases}$$

where $\mathfrak{R}_{i,j,l}$ denotes $\mathfrak{R}(\mathbf{v}_i, \mathbf{v}_j, \mathbf{l})$ and $d_{i,j}$ denotes $|n(i) - n(j)|$.

**Implications of the VLO Property.** The VLO property enforces a structured representation of video frames, ensuring that temporally adjacent frames have consistent and predictable semantic differences. When two frames have equal temporal distances from an anchor frame, their semantic gaps should be similar, fostering smooth transitions. In contrast, frames that are farther apart should exhibit larger semantic gaps, thus preserving the chronological order.

To formalize the temporal ordering constraints, we define the unique *sorted* set of frame distances from frame $i$ as $\{D_{i,1} < D_{i,2} < \cdots < D_{i,M_i}\}$, where each $D_{i,m}, m \in [M_i]$ is obtained by sorting the set $\{d_{i,j} \mid j \in [T] \backslash \{i\}\}$. Additionally, we define the count of frames at each distance level as:

$$n_{i,m} := |\{j \mid d_{i,j} = D_{i,m}, j \in [T] \backslash \{i\}\}|, \tag{4}$$

which denotes the number of frames whose temporal distance from frame $i$ equals $D_{i,m}$. The VLO property is satisfied when the proposed $\mathcal{L}_{\text{VLO}}$ approaches its theoretical lower bound, which is given by:

$$\mathcal{L}^* := \frac{1}{T(T-1)} \sum_{i=1}^{T} \sum_{m=1}^{M_i} n_{i,m} \log n_{i,m}. \tag{5}$$

This bound characterizes the optimal alignment of VL similarities, ensuring that the learned representations preserve the inherent temporal structure within the video sequence, as guaranteed by the following theorem:

**Theorem 1** (Vision-Language Ordering). *$\mathcal{L}^*$ is a tight lower bound of $\mathcal{L}_{\text{VLO}}$, i.e., $\mathcal{L}_{\text{VLO}} \geq \mathcal{L}^*$, and for any $\epsilon > 0$, there exists feature embeddings such that $\mathcal{L}_{\text{VLO}} < \mathcal{L}^* + \epsilon$. Furthermore, for any $0 < \delta < 1$, there exist $\epsilon > 0$ such that if $\mathcal{L}_{\text{VLO}} < \mathcal{L}^* + \epsilon$, the learned representations satisfy the VLO property.*

**Vision-Language Continuity.** We establish the following theoretical result to rigorously describe continuity preservation in vision-language representations:

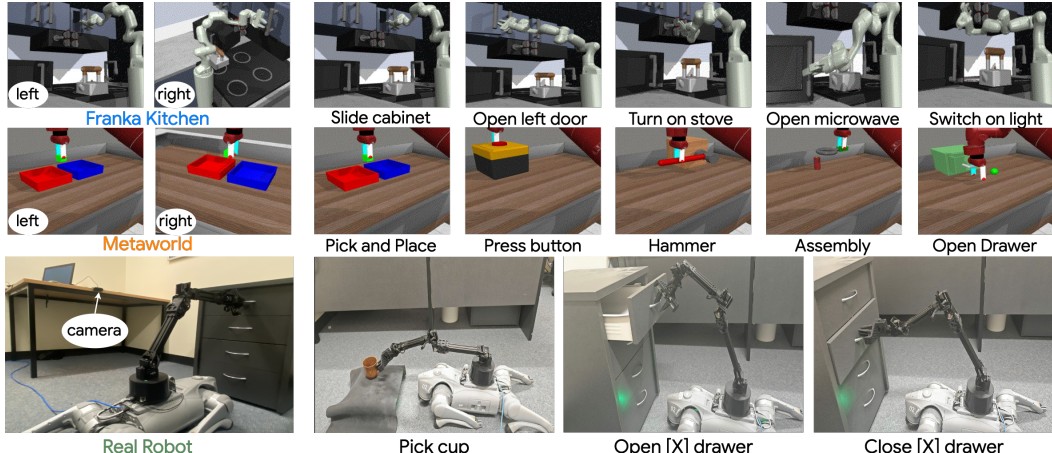

Figure 3: Policy learning environments, including 3 tasks with a real-world Unitree D1 robot arm and 5 tasks each in two simulation environments, *i.e.,* Franka Kitchen and Metaworld.

**Theorem 2** (Vision-Language Continuity). *Let $\mathbf{v}_k, \mathbf{v}_l$ be visual representations at arbitrary time steps within a Brownian Bridge-regularized interval $[n(i), n(j)]$, and let $\mathbf{l} \in \mathcal{L}$ be a language embedding. If the VL similarity function $\mathrm{sim}(\cdot)$ is Lipschitz continuous with constant $C$, then for any $\epsilon > 0$, there exists $\delta > 0$ such that:*

$$\|\mathbf{v}_k - \mathbf{v}_l\|_2 < \delta \quad \Rightarrow \quad |\mathfrak{R}(\mathbf{v}_k, \mathbf{v}_l, \mathbf{l})| < \epsilon.$$

This result follows from two key observations: (i) Brownian Bridge regularization constrains each embedding to remain close to a linear interpolation between anchor frames, with deviations governed by a time-dependent variance; and (ii) under this constraint, the distance between temporally close frames admits an explicit upper bound. Combining this with the Lipschitz continuity of the vision-language similarity function ensures that small changes in frame embeddings lead to proportionally bounded changes in alignment scores.

Building upon the continuity result, we further demonstrate that the semantic alignment score remains stable under small perturbations in language input:

**Theorem 3** (Robustness to Language Variations). *Let $\mathbf{l}'$ be a perturbed language embedding such that $\|\mathbf{l} - \mathbf{l}'\| \leq \delta_l$. Then the semantic alignment score $\mathfrak{R}$ satisfies:*

$$|\mathfrak{R}(\mathbf{v}_i, \mathbf{v}_j, \mathbf{l}') - \mathfrak{R}(\mathbf{v}_i, \mathbf{v}_j, \mathbf{l})| \leq 2C\delta_l.$$

This second result guarantees that small shifts in the language representation (*e.g.*, synonym substitution or phrasing variation) lead to bounded changes in the alignment score. Together, Theorems 2 and 3 formalize the local stability of semantic grounding across both time and modality, providing a theoretical basis for continuity-aware vision-language learning.

## 5 Experiment

In our experiments, we aim to evaluate the effectiveness of ordered and continuous vision-language representations for robotic control. First, we conduct extensive Language-Conditioned Behavior Cloning (LCBC) experiments on both real and simulated robots to validate the importance of ordering and continuity for imitation learning. Second, we assess the utility of the learned representations as reward functions on multiple real-world action videos. The results demonstrate that the ordered and continuous representations enable our method to accurately identify action boundaries and generate dense rewards aligned with the given instructions. Finally, we evaluate the robustness of our method under language perturbations, showcasing its strong generalization capability for application in real-world daily scenarios.

**Experimental Setups.** Figure 3 shows the experimental environments. For real-world robot evaluation, we deploy the **Unitree D1 robot arm** to perform three challenging manipulation tasks: pick cup, open [X] drawer and close [X] drawer, where [X] is the drawer index specified by the instruction. The pick cup task requires the model to accurately identify the cup handle, while the open/close [X] drawer tasks demand grounding of language instructions to visual observations, enabling the

Table 1: Comparison in simulation environments with varying amounts of demonstrations. Each result reports the success rate over 50 roll-outs, averaged across 2 camera views and 3 random seeds. We also report the relative performance gain in green compared to the *strongest* baseline.

| Method | FRANKA KITCHEN | | | METAWORLD | | |
|---|---|---|---|---|---|---|
| | 5 demos | 15 demos | 25 demos | 5 demos | 15 demos | 25 demos |
| CLIP | 11.67 ± 0.95 | 27.47 ± 1.01 | 31.20 ± 2.62 | 42.29 ± 2.65 | 60.33 ± 1.32 | 62.54 ± 4.36 |
| R3M | 28.60 ± 1.39 | 42.20 ± 1.00 | 51.13 ± 2.83 | 46.83 ± 3.85 | 56.50 ± 5.20 | 60.08 ± 3.62 |
| LIV | 23.40 ± 0.78 | 42.73 ± 1.17 | 51.93 ± 0.95 | 46.95 ± 2.07 | 64.33 ± 3.63 | 66.67 ± 1.49 |
| DecisionNCE | 25.33 ± 1.30 | 43.20 ± 2.25 | 50.87 ± 2.95 | 44.58 ± 2.79 | 59.08 ± 1.77 | 69.75 ± 3.90 |
| AcTOL w/o BB | 32.80 ± 1.23 | 54.20 ± 0.85 | 60.80 ± 0.87 | 50.29 ± 4.05 | 70.83 ± 4.21 | 73.33 ± 2.83 |
| AcTOL | **42.60** ± 0.53 | **61.80** ± 2.54 | **64.60** ± 0.57 | **53.81** ± 3.89 | **74.13** ± 1.59 | **81.13** ± 1.59 |
| | (+48.95%) | (+43.06%) | (+24.40%) | (+14.61%) | (+15.23%) | (+16.32%) |

model to interact with the correct drawer. To isolate manipulation performance, the Unitree Go2 quadruped remains lying down and stationary throughout the evaluation. We use a web camera to capture a third-person view as visual observation. The action space consists of a 6-DoF end-effector displacement vector and gripper state, executed at a control frequency of 20 Hz. For each task, we collect 60 demonstrations via remote control using the Unitree Go app, which is significantly fewer than the 100 trajectories typically used in prior work [26, 23]. For simulation, we choose two widely used simulation environments for evaluation: **Franka Kitchen** [16, 12] and **Metaworld** [43]. For Franka Kitchen, we evaluate five tasks: sliding a cabinet, opening the left door, opening the microwave, turning on the stove, and switching on the light. For Metaworld, we focus on learning five tasks: hammering a nail, pressing a button, picking and placing a block, assembling a ring onto a peg, and opening a drawer. Detailed environment setup can be found at Appendix B.1.

**Baselines.** Since our model is initialized with **CLIP** [34], a state-of-the-art image-text representation widely applied in various embodied tasks [9, 20, 38, 41], it is a natural choice to include CLIP as a vanilla baseline for comparison. Our primary baselines are **LIV** [26] and **DecisionNCE** [23], as we use the same model architecture and dataset for pre-training. We also compare against **R3M** [30] pre-trained on Ego4D [15], a dataset containing roughly 36× longer videos than EPIC-KITCHEN-100. We also include an ablation variant of AcTOL where the Brownian Bridge loss is removed, referred to as AcTOL w/o BB.

**Implementation Details.** We initialize our model with the weights of CLIP [34] with ResNet-50 vision backbone and further pre-train it on human action video dataset EPIC-KITCHEN-100 [10, 11]. For hyperparameter selection, we uniformly sample 10 frames of each video per batch. The loss weight $\lambda$ is 0.1. Other hyperparameters, such as temperature,s follow the default value used in CLIP [34]. More details of pre-training and hyperparameter sensitivity can be found in Appendix A.

## 5.1 Language-Conditioned Behavior Cloning

For LCBC policy learning, we keep the pre-trained vision-language encoders frozen and feed their output representations into a lightweight MLP, which is trained as a policy network.

**Simulation results.** In simulation, each task is performed from two camera viewpoints (left and right), with varying numbers of demonstrations [5, 15, 25] (*i.e.*, dataset size) for training, and evaluated under three different random seeds. We report the success rate across different environments and dataset sizes, averaged over camera views and seeds. Detailed comparison results for each task can be referred to Appendix B.5. Table 1 presents the comparison results, demonstrating that AcTOL achieves significantly enhanced performance relative to baseline methods across all evaluated datasets and environments. This superiority is particularly pronounced in the complex Franka Kitchen setting, especially under data constraints, where AcTOL with fewer demonstrations (*e.g.*, 5/15) often matches or surpasses other methods using more data (*e.g.*, 15/25), indicating its high data efficiency and robust low-resource generalization capabilities. Furthermore, ablation studies confirm the integral role of the Brownian Bridge (BB) constraint; its removal (AcTOL w/o BB) results in a significant performance decrease, validating its contribution to improving representation quality for effective policy optimization via behavior cloning.

**Robustness under visual shifts.** To further assess the ability of the model to handle visual distribution shifts, we conduct experiments in the Franka Kitchen environment following the protocol of [5]. Specifically, we compare Ac-

Figure 4: Visual shifts applied in Franka Kitchen.

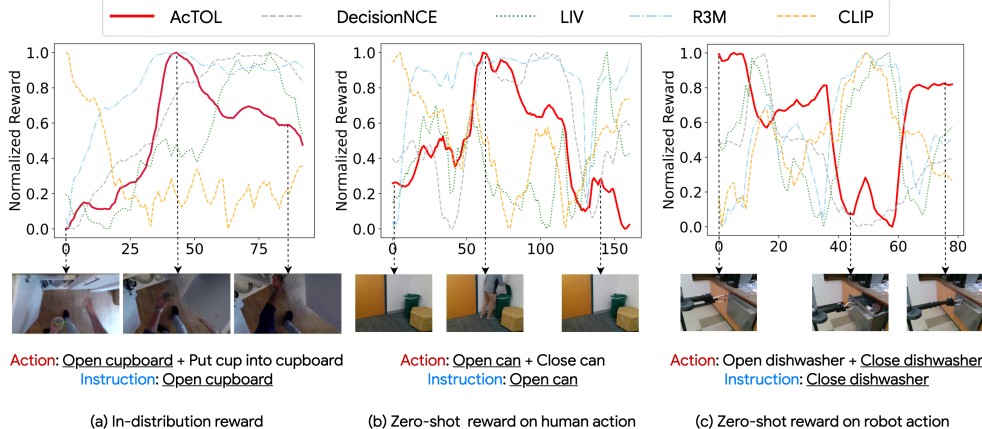

| AcTOL | DecisionNCE | LIV | R3M | CLIP |

(a) In-distribution reward    (b) Zero-shot reward on human action    (c) Zero-shot reward on robot action

Figure 5: Visualization of the normalized learned reward corresponding to different actions. Our representations effectively help capture the correct temporal order of actions in the instruction. For more results, please refer to Appendix B.6.

TOL with the strongest baseline, DecisionNCE, under visual changes absent from training. These include: (1) object distractors of increasing difficulty: easy (S1), medium (S2), and hard (S3), corresponding to scenes with 1, 3, and 9 additional YCB objects [6], respectively; and (2) background texture variations with marble hinge (S4) and metal slide (S5). All shifts are shown in Figure 4.

Policies are trained with 15 demonstrations per task, and success rates averaged over five tasks are reported in Table 2. While performance drops under visual shifts, which is expected, Ac-TOL continues to outperform DecisionNCE in all available test conditions. This suggests that the learned representation maintains useful generalization ability even without any specific adaptation for visual domain shift.

Table 2: Success rate comparison across different visual shifts in Franka Kitchen.

| Method | No shift | S1 | S2 | S3 | S4 | S5 |
|---|---|---|---|---|---|---|
| DecisionNCE | 43.2 | 27.2 | 25.6 | 4.8 | 0.0 | 8.8 |
| AcTOL | **61.8** | **43.2** | **32.8** | **9.2** | **4.4** | **38.4** |

**Real Robot results.** Table 3 shows the real robot comparison results. AcTOL consistently outperforms all baseline models across the three tasks. Among them, the pick cup task yields relatively lower performance, as it requires the model to precisely identify and grasp the cup handle, demanding stronger spatial perception capabilities. For the open/close [X] drawer tasks,

Table 3: Performance comparison on Unitree D1 arm. Success rates are reported over 10 trials.

| Method | Pick Cup | Open [X] Drawer | Close [X] Drawer |
|---|---|---|---|
| CLIP | 0 | 20 | 30 |
| R3M | 10 | 40 | 40 |
| LIV | 20 | 30 | 50 |
| DecisionNCE | 20 | 40 | 60 |
| **AcTOL** | **50** | **80** | **90** |

AcTOL is able to accurately interpret the drawer number specified in the language instruction, align it with the corresponding location in the visual observation, and execute continuous actions on the correct drawer to complete the task. These results highlight the effectiveness of AcTOL's learned visual-language representations in real-world manipulation tasks.

## 5.2 Language-Conditioned Visual Rewards

By learning semantically smooth visual representations, our model further enables the use of semantic trajectories as effective task rewards. To illustrate this, we first demonstrate the continuity of purely visual representations. In Figure 6, we visualize the learned visual representation trajectories for three tasks, each with ten video clips, using t-SNE. The results show that AcTOL significantly improves the temporal continuity of video feature trajectories while retaining CLIP's discriminative ability to distinguish

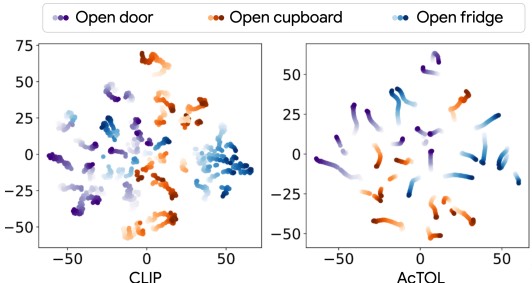

Figure 6: Visual trajectory visualization.

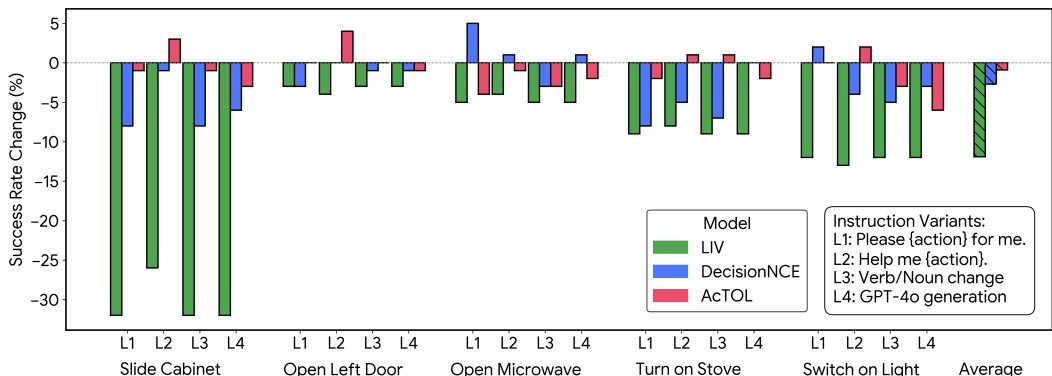

Figure 7: Success rate fluctuation across tasks in Franka Kitchen for different instruction variants.

between actions associated with different instructions. As discussed in Section 3.2, the visual continuity can stabilize learning ordered vision-language alignment. Building on this foundation, we define a dense reward signal based on the semantic alignment between the current visual state and the language goal. Specifically, at each time step $i$, we define the reward $\mathrm{cosine}(\mathbf{v}^i, \mathbf{l})$ as the similarity between the current visual state and the language goal. While prior work [26, 23] focused primarily on single-action video clips, we evaluate reward quality on three clips, each containing two consecutive actions, to assess whether the model can reliably capture fine-grained action semantics. Figure 5 (a) presents an in-distribution evaluation using a video from EPIC-KITCHEN-100. Our model produces a clear reward peak aligned with the completion of the "open cupboard" action, followed by a decline, indicating successful temporal localization of the instructed behavior. In contrast, R3M and DecisionNCE rewards continue increasing beyond the relevant action segment. Figures 5 (b) and (c) show results on real-world videos from [2], where human and robot actors perform opposite actions. Only our method consistently produces symmetric and instruction-aligned reward curves, accurately identifying both action boundaries and semantics.

## 5.3 Robustness Study under Linguistic Perturbations

In the EPIC-KITCHEN-100 dataset, textual annotations are often concise, such as ``open cupboard''. In the default setting of LCBC, we employ similarly structured simple instructions. In this experiment, to validate the robustness of the representations our method learns in real-world scenarios, we introduce several modifications to the language instructions. Specifically, we transform each original instruction into four conversational variants by varying lexical choices (*e.g.,* verbs and nouns) and incorporating ChatGPT-4o [32] generated complex instructions. Details can be found in Appendix B.4. We then evaluate the imitation learning performance conditioned on these modified instructions in the Franka Kitchen environment. For comparison, we select LIV and DecisionNCE, which are also pre-trained on EPIC-KITCHEN-100. As shown in Figure 7, the success rates of LIV and DecisionNCE dropped by 11.9% and 2.7% on average, respectively, while our method maintained a success rate comparable to that before language perturbation. This result demonstrates the robustness of our learned representations, which generalize more effectively to real-world scenarios.

## 5.4 Mitigating the Human-to-Robot Gap via Fine-Tuning

Although pretraining on human videos provides generalizable knowledge, bridging the human-to-robot domain gap remains a persistent challenge [47, 22, 31]. Since the AcTOL objectives capture the inherent temporal ordering of videos in an embodiment-agnostic manner, we can fine-tune the vision-language encoders with the same objectives used in pretraining during downstream behavior cloning. Notably, we find that fine-tuning with only a small number of robot demonstrations is sufficient to substantially mitigate the domain gap. We take 25 in-domain demonstrations (5 per task) in Franka Kitchen to fine-tune the pre-trained encoders using the AcTOL objectives. Then, as in Sec 5.1, we freeze the fine-tuned encoders and train policy networks on top using behavior cloning. We report the comparisons when using 15 demos for LCBC. As shown in Table 4, the success rate improvement demonstrates that the learned temporal inductive bias can be effectively adapted to the robot domain with limited supervision.

Table 4: Fine-tuning AcTOL encoders efficiently improves the success rate in Franka Kitchen.

| Franka Kitchen | Frozen | Finetune |
|---|---|---|
| AcTOL | 61.8 | **86.4** |

# 6 Related Work

Given the success of large-scale pre-training in the vision and language research communities [4, 24], many studies have attempted to extend this paradigm to the field of robotics. Some work leverage massive robotic trajectory data [8] for pre-training, aiming to establish unified vision-langauge-action models [48, 7, 21, 3, 13, 40, 33]. However, collecting large amounts of high-quality robot trajectory data is extremely costly and time-consuming. Consequently, many studies have begun to explore the use of large-scale, readily available, out-of-domain human action video data to learn generalizable representations that can be transferred to robotic tasks [37, 27, 35, 30, 19, 26, 28, 42, 44, 23]. Among these, TCN [37], VIP [27], MVP [35], and VC-1 [28] focus solely on studying unimodal visual representations, limiting their performance when understanding language instructions is required. R3M [30] employs language and reward models to shape progressive visual representations, while Voltron [19] and MPI [44] model the transition from the current state to the goal state conditioned on language. However, during training, these approaches freeze the language encoder, using it only to aid in the training of visual representations. As a result, they do not effectively achieve multi-modal representation learning. LIV [26] and DecisionNCE [23] have attempted to leverage CLIP [34] to train embodied multi-modal representations. LIV treats language as the goal of video actions, aligning it with the final frame, while DecisionNCE aligns language with the transition from the initial to final frame. Both rely on a goal-reaching assumption, which can lead to suboptimal results in noisy real-world videos. In contrast, our approach avoids rigid assumptions by enforcing semantic alignment that follows the intrinsic temporal continuity of videos, leading to more robust and generalizable vision-language representations. This property also benefits methods like UVD [46], which rely on pretrained visual features to detect phase changes and decompose long-horizon tasks. Our method more reliably identifies action phases, enabling stronger progress rewards and improving suitability for such goal-conditioned downstream tasks.

# 7 Conclusion and Limitations

We present Action Temporal Coherence Learning (AcTOL) as a promising vision-language pre-training solution for generalizable embodied agents. By learning action consistency from a large corpus of human action videos, AcTOL theoretically ensures the ordering and continuity of vision-language representations, as well as robustness to language perturbations. Extensive experiments across various environments demonstrate that AcTOL effectively generalizes to complex robotic manipulation tasks. While the temporal ordering of actions provides a strong inductive bias for many goal-directed tasks, it may not align well with tasks that involve ambiguous, repetitive, or cyclic behaviors. In such cases, the assumption of coherent progression might break down, potentially affecting the reliability of the model. Future work could explore adapting AcTOL to handle such repetitive action sequences.

## Acknowledgments and Disclosure of Funding

This work was partially supported by ARC DE240100105, DP240101814, DP230101196, DP230101753, BA24006, DE250100363, and ARC Industrial Transformation Research Hubs IH230100013.

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

# Appendix

## A  Pre-training Details

Following [26, 23], we use a modified ResNet-50 [17] from CLIP [34] for the vision encoder and a CLIP transformer for the language encoder. We initialize our model with CLIP and train them on EPIC-KITCHEN-100 [10, 11]. The training hyperparameters used during the pre-training are listed in Table 5. The training was conducted on two NVIDIA A800 GPUs taking approximately 30 hours. For hyperparameter sensitivity, we report the model performance under varying numbers of sampled frames and different values of the loss weight $\lambda$. As shown in Figure 8, increasing the number of sampled frames leads to higher success rates, likely because it better preserves the temporal ordering and continuity in the video sequence. The model shows low sensitivity to $\lambda$, as we observe that $\mathcal{L}_{BB}$ converges much faster than $\mathcal{L}_{VLO}$ due to its unimodal nature. As a result, $\mathcal{L}_{BB}$ primarily serves as a constraint during training rather than a dominant optimization objective.

Table 5: Hyper-parameters for pre-training.

| Config | Value |
| --- | --- |
| Training epochs | 1000 |
| Optimizer | Adam |
| Learning rate | $1 \times 10^{-5}$ |
| Batch size | 128 |
| Frames per video | 10 |
| Loss weight $\lambda$ | 0.1 |
| Weight decay | 0.001 |
| Momentum $(\beta_1, \beta_2)$ | 0.9, 0.999 |
| Augmentation | RandomCropResize |

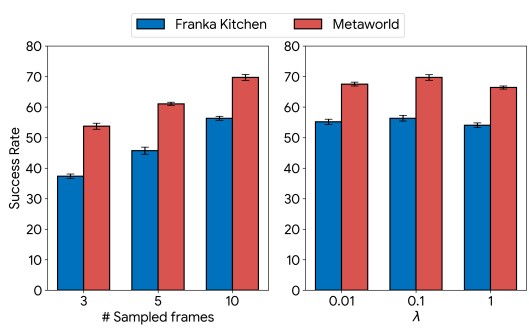

Figure 8: Hyper-parameters sensitivity.

## B  Evaluation Details

### B.1  Simulation Environment

We follow [30] for the specific simulation environment setup and code details.

**Franka Kitchen.**  The Franka Kitchen environment [16, 12] is based on the 9 degrees of freedom Franka robot. The Franka robot is placed in a kitchen environment containing several common household items: a microwave, a kettle, an overhead light, cabinets, and an oven. Following [30], the Franka Kitchen environments used in this paper are modified from their original design. Specifically, we introduce additional randomization to the scene by randomly altering the kitchen's position between episodes. This modification makes the tasks significantly more challenging in terms of both perception and control.

**Metaworld.**  The Metaworld environment [43] is an open-source simulated benchmark for robot learning. In our settings, the target object position is randomized between episodes in all tasks.

We present the specific default language instructions for each tasks in Table 6.

### B.2  Real Robot Environment

Our real robot environment is a real-world office scene where the Unitree D1 robot arm can interact with a cup and a drawer. The pick cup task requires the robot to accurately identify the handle of the cup, while the open/close [X] drawer task requires the robot to understand the drawer index specified in the language instruction and align it with the visual observation. As shown in Figure 9, we use the Unitree Go app interface to remotely control the robotic arm for action data collection. Visual observations are collected using a third-person perspective web camera in a same frequency (20Hz) with action. During control, the whole system, including AcTOL and the policy MLP, runs on a GeForce GTX 880M GPU.

Table 6: Language Instructions for tasks in Franka Kitchen and Metaworld.

| Environment ID | Language Instruction |
|---|---|
| kitchen_micro_open-v3 | open microwave |
| kitchen_sdoor_open-v3 | slide cabinet |
| kitchen_ldoor_open-v3 | open left door |
| kitchen_knob1_on-v3 | turn on stove |
| kitchen_light_on-v3 | switch on light |
| hammer-v2-goal-observable | hammer nail |
| button-press-topdown-v2-goal-observable | press button |
| bin-picking-v2-goal-observable | pick and place the block between bins |
| assembly-v2-goal-observable | assemble the ring onto peg |
| drawer-open-v2-goal-observable | open drawer |

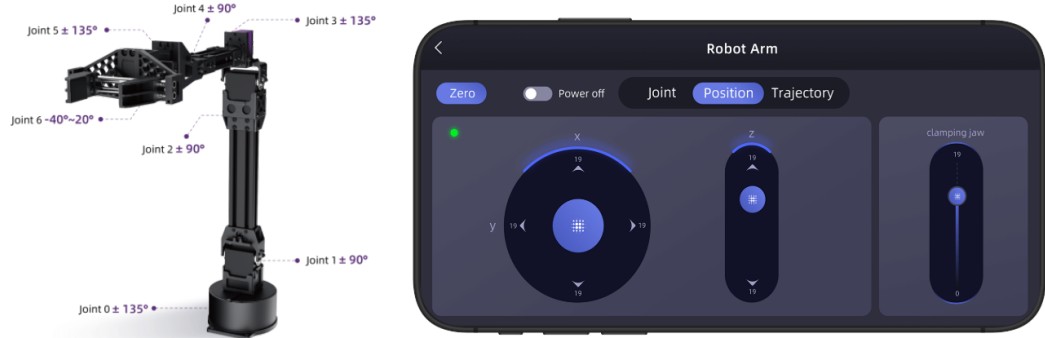

Figure 9: Action space of Unitree D1 arm and the remote control interface on Unitree Go app.

## B.3 Language-Conditioned Behavior Cloning Hyperparameters

We present the LCBC imitation learning hyperparameters in Table 7. For each distinct task in simulation, we run an evaluation episode every 1,000 gradient steps by running 50 roll-outs and computing their average success rate. Over a total of 10,000 gradient steps, we conduct this evaluation 10 times. The highest success rate among these 10 evaluations is reported as the final result. To ensure robustness, we average the results across two different camera viewpoints and three independent random seeds. In total, we run: 9 (tasks) $*2$ (views)$*3$ (demosizes) $*3$ (seeds) $*6$ (models)=972 (episodes), each episode takes approximately 2 hours on our workstation with a 24-core CPU, resulting in a total of roughly $1,944$ hours for the simulated LCBC experiments. For each task on the real robot, we use the final checkpoint and perform 10 evaluation runs with a fixed random seed, due to the cost of real-world policy evaluation.

Table 7: Hyper-parameters for LCBC.

| | Franka Kitchen | Metaworld | Real robot |
|---|---|---|---|
| MLP achitecture | [256,256] | [256,256] | [256,256] |
| Non-linear activation | ReLU | ReLU | ReLU |
| Optimizer | Adam | Adam | Adam |
| Gradient Steps | 10K | 10K | 50K |
| Learning rate | $1 \times 10^{-3}$ | $1 \times 10^{-3}$ | $1 \times 10^{-3}$ |
| Batch size | 32 | 32 | 32 |
| Horizon | 50 | 100 | 100 |
| Proprioception | 9 | 4 | No |

## B.4 Linguistic Perturbation Results

To assess the robustness of AcTOL under language perturbations, we perform extensive experiments across four instruction variants. Instructions 1 and 2 transform the original action into more con-

Table 8: Success rate fluctuation across tasks in Franka Kitchen for different instruction variants.

| Task | Instruction | LIV | DecisionNCE | AcTOL |
|------|-------------|-----|-------------|-------|
| Slide Cabinet | 1. Please slide cabinet for me. | −32 | −8 | **−1** |
| | 2. Help me slide cabinet. | −26 | −1 | **3** |
| | 3. Push open the right cupboard door. | −32 | −8 | **−1** |
| | 4. Mind pushing open the right cupboard cabinet door? I need to grab the cups inside. | −32 | −6 | **−3** |
| | **Average** | −30.5 ± 2.6 | −5.8 ± 2.9 | **−0.5 ± 2.2** |
| Open Left Door | 1. Please open left door for me. | −3 | −3 | **0** |
| | 2. Help me open left door. | −4 | 0 | **4** |
| | 3. Pull open the left cabinet door. | −3 | −1 | **0** |
| | 4. Can you pull open the left cabinet door? I need to grab something inside. | −3 | −1 | **−1** |
| | **Average** | −3.3 ± 0.4 | −1.3 ± 1.1 | **0.8 ± 1.9** |
| Open Microwave | 1. Please open microwave for me. | −5 | **5** | −4 |
| | 2. Help me open microwave. | −4 | **1** | −1 |
| | 3. Pop open the microwave oven door. | −5 | **−3** | −3 |
| | 4. Would you mind helping me pop open the microwave oven door so I can heat up my lunch? | −5 | **1** | −2 |
| | **Average** | −4.8 ± 0.4 | **1.0 ± 2.8** | −2.5 ± 1.1 |
| Turn on Stove | 1. Please turn on stove for me. | −9 | −8 | **−2** |
| | 2. Help me turn on stove. | −8 | −5 | **1** |
| | 3. Rotate the control knob to activate the stove. | −9 | −7 | **1** |
| | 4. Let us rotate the control knob to activate the stove for cooking dinner. | −9 | **0** | −2 |
| | **Average** | −8.8 ± 0.4 | −5.0 ± 3.1 | **−0.5 ± 1.5** |
| Switch on Light | 1. Please switch on light for me. | −12 | **2** | 0 |
| | 2. Help me switch on light. | −13 | −4 | **2** |
| | 3. Flip the light switch. | −12 | −5 | **−3** |
| | 4. Could you reach over and flip the light switch to brighten the kitchen area? | −12 | **−3** | −6 |
| | **Average** | −12.3 ± 0.4 | −2.5 ± 2.7 | **−1.8 ± 3.0** |
| **Average** | | −11.9 ± 0.5 | −2.7 ± 1.2 | **−0.9 ± 1.7** |

versational forms. Instruction 3 introduces vocabulary diversity by varying the verbs and nouns used. Instruction 4 further extends Instruction 3 by incorporating linguistically complex expressions generated using ChatGPT-4o. We present the comparison results obtained from experiments in the Franka Kitchen environment, with a data size of 5. As shown in Table 8, AcTOL outperforms the baselines in most instruction perturbation scenarios, thereby validating its robustness.

## B.5 Language-Conditioned Behavior Cloning Results

In Table 9- 14, we report detailed Language-Conditioned Behavior Cloning results for different task and dataset size. The results demonstrate that our method achieves significant improvements across different simulation environments, varying dataset sizes, and diverse robotic manipulation tasks.

## B.6 Language-Conditioned Visual Reward Results

As shown in Figure 10, we present more visualizations of Language-Conditioned Visual Reward on real-world robot manipulation videos from [2]. In Figure 10(a), the robot performs two consecutive and opposing actions. Our method effectively identifies the action boundaries and generates the correct reward sequence, increasing first and then decreasing, in alignment with the given instructions. In Figures 10(b)-(d), where the robot performs a single action, the robot initially moves slowly as it

Table 9: LCBC results when dataset size= 5 on Franka Kitchen.

| Method | Slide Cabinet | Open Left Door | Open Microwave | Turn On Stove | Switch On Light | Average |
|---|---|---|---|---|---|---|
| CLIP | $38.7 \pm 5.1$ | $2.0 \pm 1.0$ | $3.0 \pm 0.0$ | $7.0 \pm 2.6$ | $7.7 \pm 1.5$ | $11.7 \pm 0.9$ |
| R3M | $68.7 \pm 0.6$ | $18.3 \pm 4.0$ | $7.7 \pm 3.2$ | $19.3 \pm 7.6$ | $29.0 \pm 6.1$ | $28.6 \pm 1.4$ |
| LIV | $55.0 \pm 1.0$ | $6.0 \pm 2.9$ | $7.0 \pm 0.6$ | $13.0 \pm 0.6$ | $22.0 \pm 2.6$ | $20.6 \pm 0.7$ |
| DecisionNCE | $59.3 \pm 6.8$ | $9.7 \pm 1.5$ | $7.0 \pm 2.0$ | $\mathbf{26.3 \pm 4.5}$ | $24.3 \pm 2.5$ | $25.3 \pm 1.3$ |
| AcTOL w/o BB | $71.5 \pm 3.5$ | $11.5 \pm 0.7$ | $10.5 \pm 0.7$ | $23.5 \pm 6.4$ | $47.0 \pm 4.2$ | $32.8 \pm 2.8$ |
| AcTOL | $\mathbf{85.5 \pm 0.7}$ | $\mathbf{20.0 \pm 2.1}$ | $\mathbf{18.3 \pm 4.9}$ | $24.7 \pm 4.9$ | $\mathbf{62.3 \pm 2.8}$ | $\mathbf{42.6 \pm 0.3}$ |

Table 10: LCBC results when dataset size= 15 on Franka Kitchen.

| Method | Slide Cabinet | Open Left Door | Open Microwave | Turn On Stove | Switch On Light | Average |
|---|---|---|---|---|---|---|
| CLIP | $71.0 \pm 3.6$ | $8.0 \pm 2.0$ | $15.7 \pm 2.1$ | $14.7 \pm 0.6$ | $28.0 \pm 1.0$ | $27.5 \pm 1.0$ |
| R3M | $81.0 \pm 1.0$ | $31.0 \pm 1.7$ | $22.0 \pm 2.6$ | $19.3 \pm 4.7$ | $57.7 \pm 3.8$ | $42.2 \pm 1.0$ |
| LIV | $85.0 \pm 5.6$ | $19.0 \pm 3.0$ | $28.3 \pm 2.9$ | $29.7 \pm 3.5$ | $51.7 \pm 2.3$ | $42.7 \pm 1.2$ |
| DecisionNCE | $92.0 \pm 6.6$ | $18.7 \pm 4.5$ | $27.0 \pm 4.0$ | $33.3 \pm 3.5$ | $45.0 \pm 7.5$ | $43.2 \pm 2.3$ |
| AcTOL w/o BB | $84.5 \pm 3.5$ | $29.5 \pm 0.7$ | $29.5 \pm 2.1$ | $\mathbf{54.0 \pm 2.8}$ | $73.5 \pm 2.1$ | $54.2 \pm 0.8$ |
| AcTOL | $\mathbf{99.5 \pm 0.7}$ | $\mathbf{37.5 \pm 5.6}$ | $\mathbf{37.0 \pm 4.2}$ | $53.5 \pm 3.5$ | $\mathbf{81.5 \pm 2.1}$ | $\mathbf{61.8 \pm 2.5}$ |

Table 11: LCBC results when dataset size= 25 on Franka Kitchen.

| Method | Slide Cabinet | Open Left Door | Open Microwave | Turn On Stove | Switch On Light | Average |
|---|---|---|---|---|---|---|
| CLIP | $66.3 \pm 7.5$ | $8.7 \pm 1.2$ | $18.7 \pm 1.5$ | $23.7 \pm 3.1$ | $38.7 \pm 2.3$ | $31.2 \pm 2.6$ |
| R3M | $84.7 \pm 6.8$ | $35.3 \pm 4.0$ | $40.0 \pm 1.0$ | $34.0 \pm 5.3$ | $61.7 \pm 10.7$ | $51.1 \pm 2.8$ |
| LIV | $91.7 \pm 5.9$ | $26.0 \pm 2.6$ | $35.0 \pm 4.6$ | $45.3 \pm 0.6$ | $61.7 \pm 3.2$ | $51.9 \pm 0.9$ |
| DecisionNCE | $91.7 \pm 1.5$ | $27.0 \pm 10.4$ | $37.0 \pm 1.7$ | $47.3 \pm 1.2$ | $51.3 \pm 4.0$ | $50.9 \pm 2.9$ |
| AcTOL w/o BB | $92.0 \pm 2.4$ | $\mathbf{37.0 \pm 5.4}$ | $40.0 \pm 2.4$ | $57.0 \pm 1.5$ | $78.0 \pm 6.2$ | $60.8 \pm 1.3$ |
| AcTOL | $\mathbf{100.0 \pm 0.0}$ | $37.0 \pm 7.1$ | $\mathbf{42.5 \pm 2.1}$ | $\mathbf{62.5 \pm 2.1}$ | $\mathbf{81.0 \pm 4.2}$ | $\mathbf{64.6 \pm 0.6}$ |

Table 12: LCBC results when dataset size= 5 on Metaworld.

| Method | Assembly | Pick bin | Press button | Hammer | Open drawer | Average |
|---|---|---|---|---|---|---|
| CLIP | $48.3 \pm 5.7$ | $35.3 \pm 2.3$ | $34.3 \pm 4.9$ | $51.2 \pm 2.8$ | $91.0 \pm 1.0$ | $52.0 \pm 2.7$ |
| R3M | $63.5 \pm 5.6$ | $33.3 \pm 5.1$ | $27.3 \pm 5.1$ | $63.2 \pm 7.1$ | $92.3 \pm 0.6$ | $55.9 \pm 3.9$ |
| LIV | $61.8 \pm 6.5$ | $32.3 \pm 9.0$ | $32.7 \pm 3.5$ | $61.0 \pm 6.1$ | $\mathbf{100.0 \pm 0.0}$ | $57.7 \pm 2.1$ |
| DecisionNCE | $54.0 \pm 3.6$ | $31.0 \pm 3.6$ | $27.7 \pm 5.5$ | $65.7 \pm 3.8$ | $\mathbf{100.0 \pm 0.0}$ | $55.7 \pm 2.8$ |
| AcTOL w/o BB | $\mathbf{66.8 \pm 1.4}$ | $39.0 \pm 16.8$ | $20.7 \pm 1.5$ | $\mathbf{74.7 \pm 1.5}$ | $\mathbf{100.0 \pm 0.0}$ | $60.2 \pm 5.1$ |
| AcTOL | $62.8 \pm 6.0$ | $\mathbf{41.0 \pm 6.3}$ | $\mathbf{42.0 \pm 4.5}$ | $69.5 \pm 0.7$ | $\mathbf{100.0 \pm 0.0}$ | $\mathbf{63.1 \pm 3.9}$ |

Table 13: LCBC results when dataset size= 15 on Metaworld.

| Method | Assembly | Pick bin | Press button | Hammer | Open drawer | Average |
|---|---|---|---|---|---|---|
| CLIP | $73.0 \pm 7.8$ | $40.3 \pm 5.5$ | $52.0 \pm 7.9$ | $76.0 \pm 5.0$ | $96.7 \pm 0.6$ | $67.6 \pm 1.5$ |
| R3M | $80.7 \pm 7.6$ | $17.0 \pm 12.3$ | $45.0 \pm 4.6$ | $83.3 \pm 4.5$ | $94.0 \pm 1.0$ | $64.0 \pm 5.2$ |
| LIV | $84.3 \pm 2.5$ | $37.0 \pm 8.7$ | $54.7 \pm 3.8$ | $81.3 \pm 5.9$ | $\mathbf{100.0 \pm 0.0}$ | $71.4 \pm 3.6$ |
| DecisionNCE | $73.3 \pm 10.8$ | $36.7 \pm 5.0$ | $43.3 \pm 2.1$ | $83.0 \pm 6.0$ | $\mathbf{100.0 \pm 0.0}$ | $67.3 \pm 1.8$ |
| AcTOL w/o BB | $\mathbf{94.0 \pm 3.0}$ | $50.3 \pm 18.6$ | $48.3 \pm 1.5$ | $\mathbf{90.7 \pm 1.2}$ | $\mathbf{100.0 \pm 0.0}$ | $76.7 \pm 5.3$ |
| AcTOL | $82.5 \pm 0.7$ | $\mathbf{64.5 \pm 3.2}$ | $\mathbf{65.5 \pm 3.9}$ | $84.0 \pm 2.1$ | $\mathbf{100.0 \pm 0.0}$ | $\mathbf{79.3 \pm 1.6}$ |

Table 14: LCBC results when dataset size= 25 on Metaworld.

| Method | Assembly | Pick bin | Press button | Hammer | Open drawer | Average |
|---|---|---|---|---|---|---|
| CLIP | $69.3 \pm 5.7$ | $36.0 \pm 11.8$ | $66.0 \pm 2.5$ | $78.8 \pm 4.9$ | $99.3 \pm 0.6$ | $69.9 \pm 4.4$ |
| R3M | $87.7 \pm 2.4$ | $14.7 \pm 11.6$ | $48.3 \pm 2.1$ | $89.7 \pm 3.5$ | $\mathbf{100.0 \pm 0.0}$ | $68.1 \pm 3.6$ |
| LIV | $87.3 \pm 5.5$ | $23.7 \pm 6.8$ | $66.0 \pm 6.8$ | $89.7 \pm 2.5$ | $\mathbf{100.0 \pm 0.0}$ | $73.3 \pm 1.5$ |
| DecisionNCE | $85.7 \pm 4.9$ | $47.0 \pm 12.8$ | $58.0 \pm 7.8$ | $88.3 \pm 6.7$ | $\mathbf{100.0 \pm 0.0}$ | $75.8 \pm 3.9$ |
| AcTOL w/o BB | $\mathbf{93.7 \pm 0.6}$ | $51.7 \pm 11.9$ | $55.0 \pm 3.5$ | $\mathbf{93.0 \pm 1.0}$ | $\mathbf{100.0 \pm 0.0}$ | $78.7 \pm 3.5$ |
| AcTOL | $93.5 \pm 3.4$ | $\mathbf{66.0 \pm 2.8}$ | $\mathbf{76.5 \pm 4.9}$ | $88.5 \pm 3.9$ | $\mathbf{100.0 \pm 0.0}$ | $\mathbf{84.9 \pm 1.6}$ |

searches for the target. Correspondingly, the reward grows gradually. Once the robot interacts with the object and completes the task, our method captures the distinct semantic changes in the action, leading to a rapid reward increase. In Figures 10(e)-(f), we test two complex actions and instructions

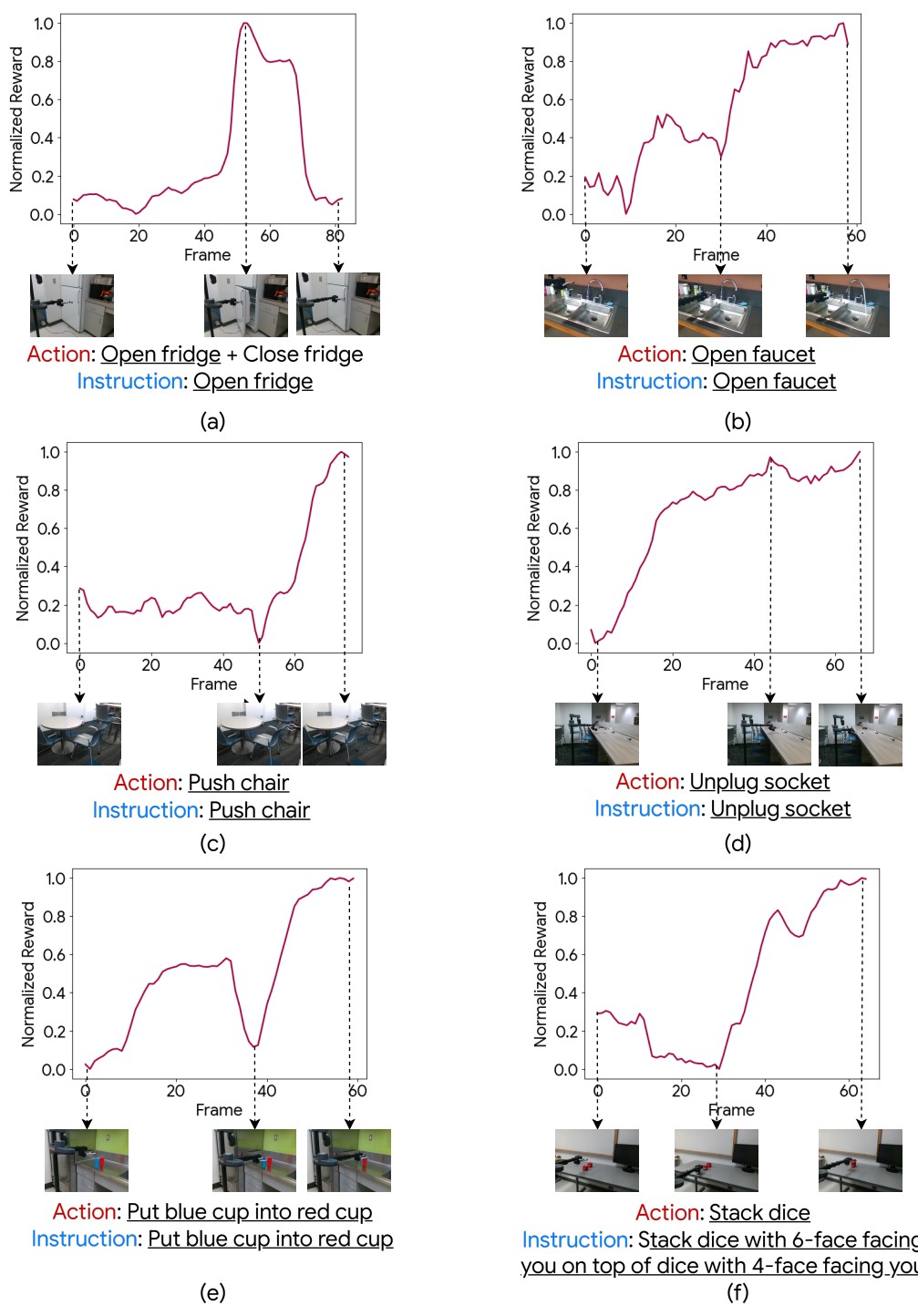

Figure 10: Reward plots for exemplar robot action videos.

to explore the limits of our method. In Figure 10(e), the model is required to accurately distinguish between the blue and red cups to complete the task. In Figure 10(f), the model needs to differentiate the orientation and face values of two dice. These scenarios impose high demands on the model's visual and semantic understanding. Our method successfully produces the correct rewards in both tasks, showcasing its potential for application in real-world, complex scenarios.

## C  Proofs

### C.1  Proofs of Theorem 1

For the proof of Theorem 1, we closely follow the approaches presented in [45] and adapted to our triplet case. We prove the theorem in three steps:

(1) $\mathcal{L}^* := \frac{1}{T(T-1)} \sum\limits_{i=1}^{T} \sum\limits_{m=1}^{M_i} n_{i,m} \log n_{i,m}$ is a lower bound of $\mathrm{L_{VLO}}$, i.e., $\mathcal{L}_{\mathrm{VLO}} > \mathcal{L}^*$.

(2) $\mathcal{L}^*$ is tight, i.e., for any $\epsilon > 0$, there exists representations such that $\mathcal{L}_{\mathrm{VLO}} < \mathcal{L}^* + \epsilon$.

(3) For any $0 < \delta < 1$, there exist $\epsilon > 0$, such that if $\mathcal{L}_{\mathrm{VLO}} < \mathcal{L}^* + \epsilon$, then the learned representations satisfy VLO property.

(1) Recall that $\mathcal{L}_{\mathrm{VLO}} = \frac{1}{T} \sum\limits_{i=1}^{T} \frac{1}{T-1} \sum\limits_{j=1,j\neq i}^{T} - \log \frac{\exp(\mathfrak{R}_{i,j,l})}{\sum\limits_{\mathbf{v}^k \in \mathcal{N}_{i,j}} \exp(\mathfrak{R}_{i,k,l})}$, where $\mathcal{N}_{i,j} = \{\mathbf{v}_k | k \neq i, d_{i,j} < d_{i,k}\}$, we rewrite it as

$$
\mathcal{L}_{\mathrm{VLO}} = -\frac{1}{T(T-1)} \sum_{i=1}^{T} \sum_{j\in[T]\setminus\{i\}} \log \frac{\exp\left(\mathfrak{R}_{i,j,l}\right)}{\sum\limits_{k\in[T]\setminus\{i\},d_{i,k}\geq d_{i,j}} \exp\left(\mathfrak{R}_{i,k,l}\right)}
$$

$$
= -\frac{1}{T(T-1)} \sum_{i=1}^{T} \sum_{m=1}^{M_i} \sum_{j\in[T]\setminus\{i\},d_{i,j}=D_{i,m}} \log \frac{\exp\left(\mathfrak{R}_{i,j,l}\right)}{\sum\limits_{k\in[T]\setminus\{i\},d_{i,k}\geq D_{i,m}} \exp\left(\mathfrak{R}_{i,k,l}\right)}
$$

$$
= -\frac{1}{T(T-1)} \sum_{i=1}^{T} \sum_{m=1}^{M_i} \sum_{j\in[T]\setminus\{i\},d_{i,j}=D_{i,m}} \log \frac{1}{\sum\limits_{k\in[T]\setminus\{i\},d_{i,k}\geq D_{i,m}} \exp\left(\mathfrak{R}_{i,k,l} - \mathfrak{R}_{i,j,l}\right)}
$$

$$
= -\frac{1}{T(T-1)} \sum_{i=1}^{T} \sum_{m=1}^{M_i} \sum_{j\in[T]\setminus\{i\},d_{i,j}=D_{i,m}} \log \frac{1}{\sum\limits_{k\in[T]\setminus\{i\},d_{i,k}=D_{i,m}} \exp\left(\mathfrak{R}_{i,k,l} - \mathfrak{R}_{i,j,l}\right)}
$$

$$
- \frac{1}{T(T-1)} \sum_{i=1}^{T} \sum_{m=1}^{M_i} \sum_{j\in[T]\setminus\{i\},d_{i,j}=D_{i,m}} \log \frac{\sum\limits_{k\in[T]\setminus\{i\},d_{i,k}=D_{i,m}} \exp\left(\mathfrak{R}_{i,k,l} - \mathfrak{R}_{i,j,l}\right)}{\sum\limits_{k\in[T]\setminus\{i\},d_{i,k}\geq D_{i,m}} \exp\left(\mathfrak{R}_{i,k,l} - \mathfrak{R}_{i,j,l}\right)}
$$

$$
= -\frac{1}{T(T-1)} \sum_{i=1}^{T} \sum_{m=1}^{M_i} \sum_{j\in[T]\setminus\{i\},d_{i,j}=D_{i,m}} \log \frac{\exp\left(\mathfrak{R}_{i,j,l}\right)}{\sum\limits_{k\in[T]\setminus\{i\},d_{i,k}=D_{i,m}} \exp\left(\mathfrak{R}_{i,k,l}\right)}
$$

$$
+ \frac{1}{T(T-1)} \sum_{i=1}^{T} \sum_{m=1}^{M_i} \sum_{j\in[T]\setminus\{i\},d_{i,j}=D_{i,m}} \log \left(1 + \frac{\sum\limits_{k\in[T]\setminus\{i\},d_{i,k}>D_{i,m}} \exp\left(\mathfrak{R}_{i,k,l} - \mathfrak{R}_{i,j,l}\right)}{\sum\limits_{k\in[T]\setminus\{i\},d_{i,k}=D_{i,m}} \exp\left(\mathfrak{R}_{i,k,l} - \mathfrak{R}_{i,j,l}\right)}\right)
$$

$$
> -\frac{1}{T(T-1)} \sum_{i=1}^{T} \sum_{m=1}^{M_i} \sum_{j\in[T]\setminus\{i\},d_{i,j}=D_{i,m}} \log \frac{\exp\left(\mathfrak{R}_{i,j,l}\right)}{\sum\limits_{k\in[T]\setminus\{i\},d_{i,k}=D_{i,m}} \exp\left(\mathfrak{R}_{i,k,l}\right)}.
$$

$$(6)$$

$\forall i \in [T], m \in [M_i]$, from Jensen's Inequality we have

$$
- \sum_{j\in[T]\setminus\{i\},d_{i,j}=D_{i,m}} \log \frac{\exp\left(\mathfrak{R}_{i,j,l}\right)}{\sum\limits_{k\in[T]\setminus\{i\},d_{i,k}=D_{i,m}} \exp\left(\mathfrak{R}_{i,k,l}\right)}
$$

$$(7)$$

$$
\geq -n_{i,m} \log \left(\frac{1}{n_{i,m}} \sum_{j\in[T]\setminus\{i\},d_{i,j}=D_{i,m}} \frac{\exp\left(\mathfrak{R}_{i,j,l}\right)}{\sum\limits_{k\in[T]\setminus\{i\},d_{i,k}=D_{i,m}} \exp\left(\mathfrak{R}_{i,k,l}\right)}\right) = n_{i,m} \log n_{i,m}.
$$

Thus, by plugging Eq. (7) into Eq. (6), we have

$$\mathcal{L}_{\text{VLO}} > \frac{1}{T(T-1)} \sum_{i=1}^{T} \sum_{m=1}^{M_i} n_{i,m} \log n_{i,m} = L^{\star} \tag{8}$$

(2) We will show for $\forall \epsilon > 0$, there is a set of representations where

$$\begin{cases} \mathfrak{R}_{i,j,l} > \mathfrak{R}_{i,k,l} + \gamma \text{ if } d_{i,j} < d_{i,k} \\ \mathfrak{R}_{i,j,l} = \mathfrak{R}_{i,k,l} \text{ if } d_{i,j} = d_{i,k} \end{cases}$$

and $\gamma := \log \frac{T}{\min\limits_{i \in [T], m \in [M_i]} n_{i,m} \epsilon}, \forall i \in [T], j, k \in [T] \backslash \{i\}$, such that $\mathcal{L}_{\text{VLO}} < L^{\star} + \epsilon$. For such a set
of representations, $\forall i \in [T], m \in [M_i], j \in \{[T] \backslash \{i\} \mid d_{i,j} = D_{i,m}\}$,

$$-\log \frac{\exp\left(\mathfrak{R}_{i,j,l}\right)}{\sum\limits_{k \in [T] \backslash \{i\}, d_{i,k} = D_{i,m}} \exp\left(\mathfrak{R}_{i,k,l}\right)} = \log n_{i,m} \tag{9}$$

since $\mathfrak{R}_{i,k,l} = \mathfrak{R}_{i,j,l}$ for all $k$ such that $d_{i,k} = D_{i,m} = d_{i,j}$, and

$$\log \left( 1 + \frac{\sum\limits_{k \in [T] \backslash \{i\}, d_{i,k} > D_{i,m}} \exp(\mathfrak{R}_{i,k,l} - \mathfrak{R}_{i,j,l})}{\sum\limits_{k \in [T] \backslash \{i\}, d_{i,k} = D_{i,m}} \exp(\mathfrak{R}_{i,k,l} - \mathfrak{R}_{i,j,l})} \right)$$
$$< \log \left( 1 + \frac{T \exp(-\gamma)}{n_{i,m}} \right) < \frac{T \exp(-\gamma)}{n_{i,m}} \leq \epsilon. \tag{10}$$

As $\mathfrak{R}_{i,k,l} - \mathfrak{R}_{i,j,l} < -\gamma$ for all $k$ such that $d_{i,k} > D_{i,m} = d_{i,j}$ and $\mathfrak{R}_{i,k,l} - \mathfrak{R}_{i,j,l} = 0$ for all $k$ such
that $d_{i,k} = D_{i,m} = d_{i,j}$. From Eq. (6) we have

$$\mathcal{L}_{\text{VLO}} = -\frac{1}{T(T-1)} \sum_{i=1}^{T} \sum_{m=1}^{M_i} \sum_{\substack{j \in [T] \backslash \{i\} \\ d_{i,j} = D_{i,m}}} \log \frac{\exp\left(\mathfrak{R}_{i,j,l}\right)}{\sum\limits_{\substack{k \in [T] \backslash \{i\} \\ d_{i,k} = D_{i,m}}} \exp\left(\mathfrak{R}_{i,k,l}\right)}$$

$$+ \frac{1}{T(T-1)} \sum_{i=1}^{T} \sum_{m=1}^{M_i} \sum_{\substack{j \in [T] \backslash \{i\} \\ d_{i,j} = D_{i,m}}} \log \left( 1 + \frac{\sum\limits_{\substack{k \in [T] \backslash \{i\} \\ d_{i,k} > D_{i,m}}} \exp\left(\mathfrak{R}_{i,k,l} - \mathfrak{R}_{i,j,l}\right)}{\sum\limits_{\substack{k \in [T] \backslash \{i\} \\ d_{i,k} = D_{i,m}}} \exp\left(\mathfrak{R}_{i,k,l} - \mathfrak{R}_{i,j,l}\right)} \right) \tag{11}$$

By plugging Eq. (9) and Eq. (10) into Eq. (11) we have

$$\mathcal{L}_{\text{VLO}} < \frac{1}{T(T-1)} \sum_{i=1}^{T} \sum_{m=1}^{M_i} n_{i,m} \log n_{i,m} + \epsilon = L^{\star} + \epsilon \tag{12}$$

(3) We will show $\forall 0 < \delta < 1$, there is a

$$\epsilon = \frac{1}{T(T-1)} \min \left( \min_{i \in [T], m \in [M_i]} \log \left( 1 + \frac{1}{n_{i,m} \exp\left(\delta + \frac{1}{\delta}\right)} \right), 2 \log \frac{1 + \exp(\delta)}{2} - \delta \right) > 0,$$

such that when $\mathcal{L}_{\text{VLO}} < L^* + \epsilon$, the representations satisfy VLO property. We first show that
$|\mathfrak{R}_{i,j,l} - \mathfrak{R}_{i,k,l}| < \delta$ if $d_{i,j} = d_{i,k}$, $i \in [T], j, k \in [T] \backslash \{i\}$ when $L_{\text{VLO}} < L^* + \epsilon$. From Eq. (6) we
have

$$\mathcal{L}_{\text{VLO}} > -\frac{1}{T(T-1)} \sum_{i=1}^{T} \sum_{m=1}^{M_i} \sum_{j \in [T] \setminus \{i\}, d_{i,j} = D_{i,m}} \log \frac{\exp(\mathfrak{R}_{i,j,l})}{\sum\limits_{k \in [T] \setminus \{i\}, d_{i,k} - D_{i,m}} \exp(\mathfrak{R}_{i,k,l})} \qquad (13)$$

Let $p_{i,m} := \underset{j \in [T] \setminus \{i\}, d_{i,j} = D_{i,m}}{\arg\min} \mathfrak{R}_{i,j,l}, q_{i,m} := \underset{j \in [T] \setminus \{i\}, d_{i,j} = D_{i,m}}{\arg\max} \mathfrak{R}_{i,j,l}, \zeta_{i,m} := \mathfrak{R}_{i,p_{i,m},l}, \eta_{i,m} :=$
$s_{i,q_{i,m},l} - s_{i,p_{i,m},l}, \forall i \in [T], m \in [M_i]$, by splitting out the maximum term and the minimum term
we have

$$\mathcal{L}_{\text{VLO}} > -\frac{1}{T(T-1)} \sum_{i=1}^{T} \sum_{m=1}^{M_i} \left\{ \log \frac{\exp(\zeta_{i,m})}{\sum\limits_{k \in [T] \setminus \{i\}, d_{i,i} = D_{i,m}} \exp(\mathfrak{R}_{i,k,l})} \right.$$

$$\left. + \log \frac{\exp(\zeta_{i,m} + \eta_{i,m})}{\sum\limits_{k \in [T] \setminus \{i\}, d_{i,k} = D_{i,m}} \exp(\mathfrak{R}_{i,k,l})} + \log \frac{\exp\left(\sum\limits_{j \in [T] \setminus \{i,p_{i,m},q_{i,m}\}, d_{i,j} = D_{i,m}} \mathfrak{R}_{i,j,l}\right)}{\left(\sum\limits_{k \in [T] \setminus \{i\}, d_{i,k} = D_{i,m}} \exp(\mathfrak{R}_{i,k,l})\right)^{n_{i,m}-2}} \right\}. \qquad (14)$$

Let $\theta_{i,m} := \frac{1}{n_{i,m}-2} \sum\limits_{j \in [T] \setminus \{i,p_{i,m},q_{i,m}\}, d_{i,j} = D_{i,j}} \exp(\mathfrak{R}_{i,j,l} - \zeta_{i,m})$, we have

$$-\log \frac{\exp(\zeta_{i,m})}{\sum\limits_{k \in [T] \setminus \{i\}, d_{i,k} = D_{i,m}} \exp(\mathfrak{R}_{i,k,l})} = \log(1 + \exp(\eta_{i,m}) + (n_{i,m} - 2)\theta_{i,m}) \qquad (15)$$

and

$$-\log \frac{\exp(\zeta_{i,m} + \eta_{i,m})}{\sum\limits_{k \in [T] \setminus \{i\}, d_{i,k} = D_{i,m}} \exp(\mathfrak{R}_{i,k,l})} = \log(1 + \exp(\eta_{i,m}) + (n_{i,m} - 2)\theta_{i,m}) - \eta_{i,m} \qquad (16)$$

Then, from Jensen's inequality, we know

$$\exp\left(\sum_{\substack{j \in [T] \setminus \{i,p_{i,m},q_{i,m}\} \\ d_{i,j} = D_{i,m}}} \mathfrak{R}_{i,j,l}\right) \leq \left(\frac{1}{n_{i,m}-2} \sum_{\substack{j \in [T] \setminus \{i,p_{i,m},q_{i,m}\} \\ d_{i,j} = D_{i,m}}} \exp(\mathfrak{R}_{i,j,l})\right)^{n_{i,m}-2} \qquad (17)$$

thus

$$-\log \frac{\exp\left(\sum_{\substack{j \in [T] \setminus \{i,p_{i,m},q_{i,m}\} \\ d_{i,j} = D_{i,m}}} \mathfrak{R}_{i,j,l}\right)}{\left(\sum_{\substack{k \in [T] \setminus \{i\} \\ d_{i,k} = D_{i,m}}} \exp(\mathfrak{R}_{i,k,l})\right)^{n_{i,m}-2}}$$

$$\geq (n_{i,m} - 2)\log(1 + \exp(\eta_{i,m}) + (n_{i,m} - 2)\theta_{i,m}) - (n_{i,m} - 2)\log(\theta_{i,m}) \qquad (18)$$

By plugging Eq. (15), Eq. (16) and Eq. (18) into Eq. (14), we have

$$\mathcal{L}_{\text{VLO}} > \frac{1}{T(T-1)} \sum_{i=1}^{T} \sum_{m=1}^{M_i} \Big( n_{i,m} \log(1 + \exp(\eta_{i,m}) + (n_{i,m} - 2)\theta_{i,m})$$

$$- \eta_{i,m} - (n_{i,m} - 2)\log(\theta_{i,m}) \Big) \qquad (19)$$

Let $h(\theta) := n_{i,m} \log\left(1 + \exp\left(\eta_{i,m}\right) + \left(n_{i,m} - 2\right)\theta\right) - \eta_{i,m} - \left(n_{i,m} - 2\right)\log(\theta)$. From derivative analysis we know $h(\theta)$ decreases monotonically when $\theta \in \left[1, \frac{1+\exp(\eta_{i,m})}{2}\right]$ and increases monotonically when $\theta \in \left[\frac{1+\exp(\eta_{i,m})}{2}, \exp\left(\eta_{i,m}\right)\right]$, thus

$$h(\theta) \geq h\left(\frac{1 + \exp\left(\eta_{i,m}\right)}{2}\right) = n_{i,m}\log n_{i,m} + 2\log\frac{1 + \exp\left(\eta_{i,m}\right)}{2} - \eta_{i,m}. \tag{20}$$

By plugging Eq. (20) into Eq. (19), we have

$$\mathcal{L}_{\mathrm{VLO}} > \frac{1}{T(T-1)}\sum_{i=1}^{T}\sum_{m=1}^{M_i}\left(n_{i,m}\log n_{i,m} + 2\log\frac{1 + \exp\left(\eta_{i,m}\right)}{2} - \eta_{i,m}\right)$$
$$= L^\star + \frac{1}{T(T-1)}\sum_{i=1}^{T}\sum_{m=1}^{M_i}\left(2\log\frac{1 + \exp\left(\eta_{i,m}\right)}{2} - \eta_{i,m}\right) \tag{21}$$

Then, since $\eta_{i,m} \geq 0$, we have $2\log\frac{1+\exp(\eta_{i,m})}{2} - \eta_{i,m} \geq 0$. Thus, $\forall i \in [T], m \in [M_i]$,

$$\mathcal{L}_{\mathrm{VLO}} > L^\star + \frac{1}{T(T-1)}\left(2\log\frac{1 + \exp\left(\eta_{i,m}\right)}{2} - \eta_{i,m}\right) \tag{22}$$

If $\mathcal{L}_{\mathrm{VLO}} < L^\star + \epsilon \leq L^\star + \frac{1}{T(T-1)}\left(2\log\frac{1+\exp(\delta)}{2} - \delta\right)$, then

$$2\log\frac{1 + \exp\left(\eta_{i,m}\right)}{2} - \eta_{i,m} < 2\log\frac{1 + \exp(\delta)}{2} - \delta \tag{23}$$

Since $y(x) = 2\log\frac{1+\exp(x)}{2} - x$ increases monotonically when $x > 0$, we have $\eta_{i,m} < \delta$. Hence $\forall i \in [T], j, k \in [T]\setminus\{i\}$, if $d_{i,j} = d_{i,k} = D_{i,m}, |\mathfrak{R}_{i,j,l} - \mathfrak{R}_{i,k,l}| \leq \eta_{i,m} < \delta$. Next, we show $\mathfrak{R}_{i,j,l} > \mathfrak{R}_{i,k,l} + \delta$ if $d_{i,j} < d_{i,k}$ when $\mathcal{L}_{\mathrm{VLO}} < L^\star + \epsilon$. From Eq. (6) we have

$$\mathcal{L}_{\mathrm{VLO}} = -\frac{1}{T(T-1)}\sum_{i=1}^{T}\sum_{m=1}^{M_i}\sum_{\substack{j\in[T]\setminus\{i\}\\ d_{i,j}=D_{i,m}}}\log\frac{\exp(\mathfrak{R}_{i,j,l})}{\sum_{\substack{k\in[T]\setminus\{i\}\\ d_{i,k}=D_{i,m}}}\exp(\mathfrak{R}_{i,k,l})}$$
$$+ \frac{1}{T(T-1)}\sum_{i=1}^{T}\sum_{m=1}^{M_i}\sum_{\substack{j\in[T]\setminus\{i\}\\ d_{i,j}=D_{i,m}}}\log\left(1 + \frac{\sum_{\substack{k\in[T]\setminus\{i\}\\ d_{i,k}=D_{i,m}}}\exp(\mathfrak{R}_{i,k,l}-\mathfrak{R}_{i,j,l})}{\sum_{\substack{k\in[T]\setminus\{i\}\\ d_{i,k}>D_{i,m}}}\exp(\mathfrak{R}_{i,k,l}-\mathfrak{R}_{i,j,l})}\right) \tag{24}$$

and combining it with Eq. (7) we have

$$\mathcal{L}_{\mathrm{VLO}} \geq L^\star + \frac{1}{T(T-1)}\sum_{i=1}^{T}\sum_{m=1}^{M_i}\sum_{\substack{j\in[T]\setminus\{i\}\\ d_{i,j}=D_{i,m}}}\log\left(1 + \frac{\sum_{\substack{k\in[T]\setminus\{i\}\\ d_{i,k}>D_{i,m}}}\exp(\mathfrak{R}_{i,k,l}-\mathfrak{R}_{i,j,l})}{\sum_{\substack{k\in[T]\setminus\{i\}\\ d_{i,k}=D_{i,m}}}\exp(\mathfrak{R}_{i,k,l}-\mathfrak{R}_{i,j,l})}\right)$$
$$> L^\star + \frac{1}{T(T-1)}\log\left(1 + \frac{\exp(\mathfrak{R}_{i,k,l}-\mathfrak{R}_{i,j,l})}{\sum_{\substack{h\in[T]\setminus\{i\}\\ d_{i,h}=d_{i,j}}}\exp(\mathfrak{R}_{i,h,l}-\mathfrak{R}_{i,j,l})}\right) \tag{25}$$

$\forall i \in [T], j \in [T]\backslash\{i\}, k \in \{k \in [T]\backslash\{i\} \mid d_{i,j} < d_{i,k}\}$. When $\mathcal{L}_{\text{VLO}} < L^\star + \epsilon$, we already have $|\mathfrak{R}_{i,h,l} - \mathfrak{R}_{i,j,l}| < \delta, \forall d_{i,h} = d_{i,j}$, which derives $\mathfrak{R}_{i,h,l} - \mathfrak{R}_{i,j,l} < \delta$ and thus $\exp(\mathfrak{R}_{i,h,l} - \mathfrak{R}_{i,j,l}) < \exp(\delta)$. By putting this into Eq. (24), we have $\forall i \in [T], j \in [T]\backslash\{i\}, k \in \{k \in [T]\backslash\{i\} \mid d_{i,j} < d_{i,k}\}$,

$$\mathcal{L}_{\text{VLO}} > L^\star + \frac{1}{T(T-1)} \log\left(1 + \frac{\exp(\mathfrak{R}_{i,k,l} - \mathfrak{R}_{i,j,l})}{n_{i,r_{i,j}} \exp(\delta)}\right) \tag{26}$$

where $r_{i,j} \in [M_i]$ is the index such that $D_{i,r_{i,j}} = d_{i,j}$.

Further, given $\mathcal{L}_{\text{VLO}} < L^\star + \epsilon < L^\star + \frac{1}{T(T-1)} \log\left(1 + \frac{1}{n_{i,r_{i,j}} \exp\left(\delta + \frac{1}{\delta}\right)}\right)$, we have

$$\log\left(1 + \frac{\exp(\mathfrak{R}_{i,k,l} - \mathfrak{R}_{i,j,l})}{n_{i,r_{i,j}} \exp(\delta)}\right) < \log\left(1 + \frac{1}{n_{i,r_{i,j}} \exp\left(\delta + \frac{1}{\delta}\right)}\right) \tag{27}$$

which derives $\mathfrak{R}_{i,j,l} > \mathfrak{R}_{i,k,l} + \frac{1}{\delta}, \forall i \in [T], j \in [T]\backslash\{i\}, k \in \{[T]\backslash\{i\} \mid d_{i,j} < d_{i,k}\}$. Finally, $\forall i \in [T], j, k \in [T]\backslash\{i\}, \mathfrak{R}_{i,j,l} < \mathfrak{R}_{i,k,l} - \frac{1}{\delta}$ if $d_{i,j} > d_{i,k}$ directly follows from $\mathfrak{R}_{i,j,l} > \mathfrak{R}_{i,k,l} + \frac{1}{\delta}$ if $d_{i,j} < d_{i,k}$. $\square$

## C.2 Proofs of Theorem 2

**Setup and Assumptions.** To provide the vision-language continuity, we first assume that the frame embeddings $\{\mathbf{v}_t\}$, where $t \in [1, T]$ are regularized under a Brownian Bridge process $\mathbf{B}(t)$ as discussed in Section 3.2, where the transition density for any intermediate time $t \in [n(i), n(j)]$ within a sampled interval is given as:

$$\mathbf{B}(t) \sim \mathcal{N}\left(\mathbb{E}[\mathbf{B}(t)], \text{Var}[\mathbf{B}(t)]\right), \tag{28}$$

with:

$$\mathbb{E}[\mathbf{B}(t)] = \mathbf{v}_i + \frac{t - n(i)}{n(j) - n(i)}(\mathbf{v}_j - \mathbf{v}_i), \ \text{Var}[\mathbf{B}(t)] = \frac{(t - n(i))(n(j) - t)}{n(j) - n(i)}. \tag{29}$$

All time steps $t \in [1, T]$ are covered by at least one sampled interval, ensuring the entire video sequence satisfies the Brownian Bridge regularization. Now, let $\mathbf{v}_k, \mathbf{v}_l \in \mathbb{R}^d$ be arbitrary embeddings, *not necessarily* the endpoints $\mathbf{v}_i$ and $\mathbf{v}_j$ of a sampled interval. These embeddings fall within the *union* $\mathfrak{U}$ of all sampled local intervals. Without loss of generality, here we can identify the interval $[n(i), n(j)] \in \mathfrak{U}$ from the union containing $\mathbf{v}_k$ and $\mathbf{v}_l$.

**Bounding Local Continuity.** Recall that semantic alignment score $\mathfrak{R}(\mathbf{v}_k, \mathbf{v}_l, \mathbf{l})$ is defined as:

$$\mathfrak{R}(\mathbf{v}_k, \mathbf{v}_l, \mathbf{l}) = -\|\text{sim}(\mathbf{v}_k, \mathbf{l}) - \text{sim}(\mathbf{v}_l, \mathbf{l})\|_2,$$

where $\text{sim}(\cdot)$ is Lipschitz continuous with constant $C > 0$ when embeddings are normalized as unit vectors. By the Lipschitz continuity of $\text{sim}(\cdot)$, we have:

$$\|\text{sim}(\mathbf{v}_k, \mathbf{l}) - \text{sim}(\mathbf{v}_l, \mathbf{l})\|_2 \leq C \cdot \|\mathbf{v}_k - \mathbf{v}_l\|_2.$$

To ensure the continuity of $\mathfrak{R}$, we must bound $\|\mathbf{v}_k - \mathbf{v}_l\|_2$. Under the Brownian Bridge regularization, the embeddings are aligned with the mean trajectory $\mathbb{E}[\mathbf{B}(t)]$, and deviations are constrained by the variance $\text{Var}[\mathbf{B}(t)]$. Specifically:

$$\|\mathbf{v}_t - \mathbb{E}[\mathbf{B}(t)]\|_2^2 \leq \lambda \cdot \text{Var}[\mathbf{B}(t)],$$

where $\lambda > 0$ depends on the strength of the Brownian Bridge loss $\mathcal{L}_{\text{BB}}$. Below we omit $\lambda$ for simplicty. Substituting the variance:

$$\text{Var}[\mathbf{B}(t)] = \frac{(t - n(i))(n(j) - t)}{n(j) - n(i)}.$$

**Bounding Pairwise Distance.** The total pairwise distance between $\mathbf{v}_k$ and $\mathbf{v}_l$ can be expressed as:

$$\|\mathbf{v}_k - \mathbf{v}_l\|_2 \leq \|\mathbb{E}[\mathbf{B}(k)] - \mathbb{E}[\mathbf{B}(l)]\|_2 + \sqrt{\text{Var}[\mathbf{B}(k)]} + \sqrt{\text{Var}[\mathbf{B}(l)]}.$$

Since the mean trajectory $\mathbb{E}[\mathbf{B}(t)]$ is linear within the interval $[n(i), n(j)]$, we have:

$$\|\mathbb{E}[\mathbf{B}(k)] - \mathbb{E}[\mathbf{B}(l)]\|_2 \leq \frac{|k - l|}{n(j) - n(i)}\|\mathbf{v}_j - \mathbf{v}_i\|_2.$$

Combining these bounds, now we can rewrite into the following inequality:

$$\|\mathbf{v}_k - \mathbf{v}_l\|_2 \leq \frac{|k - l|}{n(j) - n(i)}\|\mathbf{v}_j - \mathbf{v}_i\|_2 + \sqrt{\frac{(k - n(i))(n(j) - k)}{n(j) - n(i)}} + \sqrt{\frac{(l - n(i))(n(j) - l)}{n(j) - n(i)}}.$$

For the variance terms, the Brownian Bridge process achieves its maximum variance at the midpoint $t = \frac{n(i)+n(j)}{2}$. This gives us,

$$\text{Var}[\mathbf{B}(t_{\max})] = \frac{n(j) - n(i)}{4}, \quad \|\mathbf{v}_k - \mathbf{v}_l\|_2 \leq 2\frac{|k - l|}{n(j) - n(i)} + \sqrt{(n(j) - n(i))}.$$

**Bounding Semantic Alignment Score.** Finally, by substituting this bound into the Lipschitz continuity of sim, we obtain,

$$|\mathfrak{R}(\mathbf{v}_k, \mathbf{v}_l, \mathbf{l})| \leq C \cdot \left(\frac{2|k - l|}{n(j) - n(i)} + \sqrt{(n(j) - n(i))}\right).$$

To ensure $|\mathfrak{R}(\mathbf{v}_k, \mathbf{v}_l, \mathbf{l})| < \epsilon$, we require:

$$C \cdot \left(2\frac{|k - l|}{n(j) - n(i)} + \sqrt{n(j) - n(i)}\right) < \epsilon.$$

Here, we consider these two terms respectively:

$$2C\frac{|k - l|}{n(j) - n(i)} < \frac{\epsilon}{2}, \quad C\sqrt{n(j) - n(i)} < \frac{\epsilon}{2},$$

which gives:

$$|k - l| < \delta_1 = \frac{\epsilon \cdot (n(j) - n(i))}{4C}, \quad n(j) - n(i) < \left(\frac{\epsilon}{2C}\right)^2.$$

Combining these conditions, we choose:

$$\delta = \min\left(\frac{\epsilon \cdot (n(j) - n(i))}{4C}, \frac{\epsilon^2}{4C^2}\right).$$

**Final Conclusion.** For any given $\epsilon > 0$, setting $\delta = \min\left(\frac{\epsilon \cdot (n(j) - n(i))}{4C}, \frac{\epsilon^2}{4C^2}\right)$ ensures:

$$\|\mathbf{v}_k - \mathbf{v}_l\|_2 < \delta \quad \Rightarrow \quad |\mathfrak{R}(\mathbf{v}_k, \mathbf{v}_l, \mathbf{l})| < \epsilon.$$

$\square$

### C.3 Proofs of Theorem 3

From the definition of the semantic alignment score, we have:

$$\mathfrak{R}(\mathbf{v}_i, \mathbf{v}_j, \mathbf{l}) = -|\text{sim}(\mathbf{v}_i, \mathbf{l}) - \text{sim}(\mathbf{v}_j, \mathbf{l})|, \quad \mathfrak{R}(\mathbf{v}_i, \mathbf{v}_j, \mathbf{l}') = -|\text{sim}(\mathbf{v}_i, \mathbf{l}') - \text{sim}(\mathbf{v}_j, \mathbf{l}')|.$$

The difference in scores can be bounded using the reverse triangle inequality:

$$|\mathfrak{R}(\mathbf{v}_i, \mathbf{v}_j, \mathbf{l}') - \mathfrak{R}(\mathbf{v}_i, \mathbf{v}_j, \mathbf{l})| \leq |(\text{sim}(\mathbf{v}_i, \mathbf{l}') - \text{sim}(\mathbf{v}_j, \mathbf{l}')) - (\text{sim}(\mathbf{v}_i, \mathbf{l}) - \text{sim}(\mathbf{v}_j, \mathbf{l}))|.$$

Simplifying the inequalities above, it gives us:

$$|\mathfrak{R}(\mathbf{v}_i, \mathbf{v}_j, \mathbf{l}') - \mathfrak{R}(\mathbf{v}_i, \mathbf{v}_j, \mathbf{l})| \leq |\text{sim}(\mathbf{v}_i, \mathbf{l}') - \text{sim}(\mathbf{v}_i, \mathbf{l})| + |\text{sim}(\mathbf{v}_j, \mathbf{l}') - \text{sim}(\mathbf{v}_j, \mathbf{l})|.$$

By the Lipschitz continuity of sim, we have: for some constant $C > 0$,

$$|\text{sim}(\mathbf{v}_i, \mathbf{l}') - \text{sim}(\mathbf{v}_i, \mathbf{l})| \leq C\|\mathbf{l}' - \mathbf{l}\|_2, \quad |\text{sim}(\mathbf{v}_j, \mathbf{l}') - \text{sim}(\mathbf{v}_j, \mathbf{l})| \leq C\|\mathbf{l}' - \mathbf{l}\|_2.$$

Substituting these bounds and considering $\|\mathbf{l}' - \mathbf{l}\|_2 \leq \delta_l$

$$|\mathfrak{R}(\mathbf{v}_i, \mathbf{v}_j, \mathbf{l}') - \mathfrak{R}(\mathbf{v}_i, \mathbf{v}_j, \mathbf{l})| \leq 2C\|\mathbf{l}' - \mathbf{l}\|_2 \leq 2C\delta_l. \tag{30}$$

$\square$

## D   Broader Impacts

We introduce Action Temporal Coherence Learning (AcTOL), a vision-language pretraining framework aimed at improving the generalization capabilities of embodied agents in a variety of manipulation tasks. By learning from large-scale human action videos, AcTOL helps agents acquire temporally consistent representations aligned with natural language, which can support more flexible and data-efficient robotic learning. However, some potential risks should be acknowledged. If AcTOL is trained on video data that contains societal biases or stereotypes, those patterns may be reflected in the model's behavior. For instance, if certain groups or actions are underrepresented or portrayed inaccurately, the resulting agents could behave in ways that are inappropriate or unreliable in diverse real-world settings. While these challenges are common across many data-driven systems in robotics and vision-language learning, we believe future work should explore strategies such as dataset auditing, fairness-aware training, and improved transparency to support more responsible and robust deployment.

