# OpenReview forum: "Provable Ordering and Continuity in Vision-Language Pretraining for Generalizable Embodied Agents"
_NeurIPS.cc/2025/Conference — NeurIPS 2025 poster_

### Official Review · Reviewer_LtsP · 2025-06-22

**Clarity:** 3
**Significance:** 3
**Originality:** 3
**Rating:** 5
**Confidence:** 4

**Summary:**

This paper introduces Action Temporal Coherence Learning (AcTOL), a novel vision-language pre-training framework for generalizable embodied agents. Addressing limitations of prior time contrastive learning methods that overemphasize goal-reaching heuristics, AcTOL learns ordered and continuous vision-language representations without rigid goal-based constraints. It achieves this by simultaneously contrasting semantic differences between video frames to reflect their natural ordering and imposing a local Brownian bridge constraint to ensure smooth transitions across intermediate frames. Extensive imitation learning experiments on simulated (Meta-World, Franka Kitchen) and real (Mobile ALOHA) robots demonstrate that AcTOL's pretrained features significantly enhance downstream manipulation tasks, showing robustness to linguistic styles and offering a viable pathway to generalized embodied agents.

**Questions:**

Briefly discussing the potential applicability of AcTOL's learned features for other embodied AI tasks (e.g., navigation, planning) beyond manipulation would broaden its perceived impact.

**Ethical Concerns:**

["NO or VERY MINOR ethics concerns only"]

**Final Justification:**

I have read the author response and found that it addresses the concerns I previously raised. There are no remaining major issues from my perspective. I will keep my original rating.

**Limitations:**

yes

**Quality:**

4

**Strengths And Weaknesses:**

Strengths
-  The paper clearly identifies and addresses a critical limitation of existing vision-language pre-training methods for embodied agents, specifically the "overemphasis on future frames" and its detrimental effects on learned representations. The problem of learning intrinsic temporal coherence (ordering and continuity) without rigid goal-based constraints is well-motivated.
-  AcTOL proposes a genuinely novel approach by combining two distinct yet complementary learning objectives: semantic contrast for temporal ordering and a local Brownian bridge constraint for continuity.
-  Experiments are extensive, spanning diverse simulated environments (Meta-World, Franka Kitchen) and a real-world robot (Mobile ALOHA), which greatly enhances the generalizability claims. The method significantly outperforms strong, recent baselines in embodied AI VLP.
-  The paper is exceptionally well-written, clear, and easy to follow.

Weaknesses
- While the title claims "Provable Ordering and Continuity," and Appendix A provides mathematical theorems, the connection between these formal proofs and the practical implications or guarantees on the learned representations in the main text could be stronger. The proofs are primarily existence/property proofs rather than guarantees of real-world "provability" in the context of learned features. More discussion on this link would strengthen the claim.
- While the "local" Brownian bridge constraint is a clever way to handle continuity, the paper doesn't explicitly discuss how AcTOL would scale to extremely long or hierarchical video sequences, where global temporal coherence might become more challenging to model purely through local constraints. The computational cost of pairwise comparisons for L_order might also increase significantly with very long sequences.

---

> ### Author Rebuttal · Authors · 2025-07-29
>
> Thank you for your thoughtful feedback! Below, we provide point-by-point responses to each of your questions.
> > **W1. Weak connection between theoretical proofs and practical guarantees.**
>
> Thanks for your suggestions. Our theoretical analysis aims to offer insight into how our self-supervised objectives induce ordered and continuous representations, moving beyond traditional goal-reaching assumptions. While these proofs do not provide direct guarantees on model behavior in practical deployments, **our empirical results demonstrate that the theoretical properties are indeed reflected in real-world performance**.
>
> Specifically, **Theorems 1 and 2** establish that the learned feature space is ordered and continuous, which directly supports real-world imitation learning by enabling more **accurate and data-efficient behavior cloning** in continuous action spaces. This structure also supports generating **dense progress rewards** that can effectively identify task boundaries, benefiting real-world reinforcement learning. Additionally, **Theorem 3** guarantees robustness to linguistic perturbations, an important property for **real-world robotic applications involving natural language instructions**.  We will further clarify these connections and provide additional discussions in the revised version.
>
> > **W2. Scalability to long or hierarchical sequences.**
>
> Thanks for raising this meaningful concern. The global temporal coherence in AcTOL is primarily maintained through the Vision-Language Ordering (VLO) loss applied during pretraining. This loss samples frames randomly from each video within a batch and encourages consistent temporal ordering. For very long videos, it is true that more batches may be required to cover most frame pairs. However, it is **neither necessary nor efficient to optimize over every possible frame pair**. Instead, training should be viewed as optimizing the **expected** VLO loss across all possible random batches. Once this expectation is sufficiently minimized, Markov's inequality ensures that the loss for **any** batches will also be low enough with high probability. This statistical property allows AcTOL to scale gracefully even as sequence length grows.
>
> Regarding hierarchical or multi-task videos, such sequences can be naturally decomposed into shorter sub-trajectories using **a high-level planner**, such as **LLMs/VLMs**. Each sub-trajectory can then be modeled effectively by AcTOL using the local continuity and ordering constraints. This modular design makes AcTOL compatible with hierarchical planning and scalable to longer video sequences in practice.
>
> > **Q1. Limited discussion of broader applicability.**
>
> Thanks for the suggestion. While AcTOL is primarily demonstrated on manipulation tasks, its vision-language-aligned representations are **general** and could support other embodied AI domains such as navigation and planning. To apply AcTOL in these settings, the model would need to be **pretrained on video-instruction pairs that are specific to navigation behaviors or planning scenarios**. This ensures that the learned representations capture the appropriate domain-specific semantics and temporal patterns.
>
> For **navigation**, AcTOL could be trained on egocentric video sequences paired with spatial instructions such as “go to the sink” or “walk to the hallway.”  For **planning**, AcTOL can support high-level reasoning by evaluating the semantic progress of proposed visual plans or subgoals. We will discuss such potentials in revised version and view this as a promising direction for future exploration.

---

> > ### Comment · Reviewer_LtsP · 2025-08-01
> >
> > Thank you for your response. Your clarifications have well addressed my concerns, and I have no further questions.

---

### Official Review · Reviewer_7kXX · 2025-06-29

**Clarity:** 3
**Significance:** 3
**Originality:** 3
**Rating:** 4
**Confidence:** 3

**Summary:**

This paper introduces Action Temporal Coherence Learning (AcTOL) to learn ordered and continuous vision-language representation. AcTOL enforces vision-language similarity differences between closer frames to be smaller than those frames that are far apart. In addition, it uses Brownian bridge to ensure consistency and smoothness between the representations of intermediate frames. Results show that the proposed method learns representation that performs well in language-conditioned behavior cloning. In addition, the learned vision-language representation can be used as a reward model and is robust to linguistic perturbations.

**Questions:**

See weaknesses.

**Ethical Concerns:**

["NO or VERY MINOR ethics concerns only"]

**Final Justification:**

The rebuttal has addressed all my concerns. There are no other major concerns from my side. I raise my rating to 4.

**Limitations:**

yes

**Quality:**

3

**Strengths And Weaknesses:**

Strength:
1. AcTOL does not rely on explicit future frame selection which may be noisy when the video terminates early or include irrelevant actions. It captures the temporal semantic via ordering, emphasizing that the semantic differences between closer frames are smaller than those far apart.
2. The paper introduces Brownian bridge to address the continuity of the representations between frames.
3. The paper provides theoretical analysis and proofs to support the vision-language ordering and continuity.
4. Results show that the proposed method outperforms other representation methods in Franka Kitchen, Meteworld, and a real-robot experiment.

Weakness:
1. The paper shows that through vision-language ordering and continuity, the model is able to learn representation that can serve as a visual reward for tasks. However, I am not sure why these two conditions would lead to such outcome. The supervision for the representation seems to have no constraints on the progress towards the goal specified by the instruction. It would be great to clarify this in the paper.
2. The paper claims that AcTOL retains CLIP's capability to distinguish between actions associated with different instructions. However, the evaluation is qualitative. It would be better to perform a quantitative study on more samples to support this claim.
3. The paper perform experiments on Franka Kitchen, Metaworld, and a real robot arm with the vision-language encoder frozen. However, the zero-shot results do not include the visualization for these experiments. It would be beneficial to include qualitative visualization and quantitative analysis on the zero-shot visual reward of AcTOL on these experiments. This would help readers better understand the mechanisms that enable AcTOL to outperforms baseline methods.
4. The real-world experiment is relatively simple. It would be great to verify the zero-shot capability of AcTOL on more complex tasks. For example, is AcTOL able to handle pick-and-place of multiple objects? Is it able to generalize to unseen settings, including unseen backgrounds and objects?

---

> ### Author Rebuttal · Authors · 2025-07-29
>
> Thank you for your thoughtful feedback! Below, we provide point-by-point responses to each of your questions.
>
> > W1. Unclear justification for ordering and continuity as effective supervision.
>
> Thank you for raising this point. We attribute AcTOL’s ability to provide meaningful visual reward signals to two key factors: the **pretrained CLIP embeddings** and the **self-supervised** nature of our objectives. As shown in Figure 5, CLIP features already encode meaningful distinctions between different states in a visual trajectory, **especially the initial and goal states**. This is due to large-scale vision-language pretraining. However, these features are not naturally ordered or continuous throughout the entire trajectory.
>
> By enforcing temporal ordering and continuity, our **self-supervised objectives structure the CLIP feature space** to better reflect the progression of task execution. These constraints guide the representations to unfold in an ordered and smooth manner between the already well-separated initial and goal states，**without relying on strong goal-conditioned supervision** as in prior methods. As a result, the learned representation aligns well with task progression and can serve as a dense visual reward signal for downstream control tasks. We will clarify this point more thoroughly in the revised version of the paper.
>
> > W2. Lack of quantitative support for action-instruction alignment.
>
> Thank you for the helpful suggestion. To quantitatively assess whether AcTOL retains CLIP’s ability to distinguish between actions associated with different instructions, we evaluate the clustering quality of visual features on the EPIC-Kitchens dataset. Specifically, we selected videos from **50 different action classes**, each associated with a unique instruction. For every video, we extracted frame-level features using both CLIP and AcTOL, and aggregated them using temporal mean pooling to obtain a single feature per video. Then we adopt two widely used unsupervised clustering metrics: the **Davies–Bouldin score** (lower is better) and the **Calinski–Harabasz score** (higher is better). As shown below, AcTOL achieves better scores than CLIP on both metrics, indicating improved intra-class compactness and inter-class separation:
>
> |Methods|davies_bouldin_score($\downarrow$)|calinski_harabasz_score($\uparrow$)|
> |-|-|-|
> |AcTOL|**2.3796**|**3.4125**|
> |CLIP|2.5071|3.1580|
>
>
> These results suggest that AcTOL **preserves the discriminative structure** of CLIP representations with respect to action-instruction alignment.
>
> > W3. Missing zero-shot visualization and analysis on main benchmarks.
>
> Thank you for the helpful suggestion. Since we are currently limited to text responses during the rebuttal phase, we will include more visualizations in the revised version. To partially address this concern here, we provide a representative table below that shows the zero-shot visual reward produced by AcTOL on **our real-world robot data**.
>
> In this experiment, we evaluate the temporal evolution of visual reward for the "open first drawer" task, using two instructions: one **correct** ("Open first drawer") and one **distractor** ("Open second drawer"). The **normalized reward** is measured at different timepoints (20% to 100% of the trajectory).
>
> | Instruction | 20% | 40% | 60% | 80% | 100% |
> |-|-|-|-|-|-|
> | Open first drawer (✅)| 0.21 | 0.46 | 0.71 | 0.85 | **1.0 (peak)** |
> | Open second drawer (❌)| 0.18 | 0.59 | **1.0 (peak)** | 0.67 | 0.11 |
>
> As seen in the table, the reward for the **correct instruction** (1st row) steadily **increases as the task progresses, peaking near completion**. In contrast, the reward for **distractor instruction** (2nd row) **initially grow due to visual similarity in the early motion but diverges in later stages**, as the action clearly corresponds to a different drawer. This trend highlights AcTOL’s ability to assign fine-grained, temporally-aware rewards that distinguish between correct and incorrect tasks over time, even in a zero-shot setting without fine-tuning.
>
> > W4. Limited complexity in real-world evaluation.
>
> AcTOL is theoretically capable of handling more complex tasks; however, doing so requires **significantly more demonstrations with high quality**. As we describe in **Appendix B.2 (line 591)**, collecting demonstrations with our D1 arm is expensive because it currently relies solely on remote control. To address this limitation, we are exploring low-cost teleoperation methods to enable more scalable data collection, which would allow us to explore more diverse and complex tasks in future work.
>
> It's also worth noting that our current experiments **require fine-grained understanding of the scence layout**. For example, in the "open [X] drawer" task, we use a drawer setup with a complex layout. We collect 60 demonstrations in total, 15 for each drawer where X ∈ {"first", "second", "third", "fourth"}. During evaluation, the model is **given a randomly sampled drawer instruction** and must roll out accordingly. This setup demands fine-grained semantic understanding of the target drawer, linking language (e.g., "third drawer") to the corresponding visual context.
>
>  Additionally, we have already **added experiments under visual domain shift conditions in the Franka Kitchen environment**, please refer to **our response to Reviewer h3Cm’s W1** for details. We hope this can partially address your concern.

---

> ### Comment · Reviewer_7kXX · 2025-08-05
>
> Thank you for your response. It has addressed all my concerns. I have no further questions. I raise my rating to 4.

---

### Official Review · Reviewer_Mtcd · 2025-07-02

**Clarity:** 3
**Significance:** 3
**Originality:** 3
**Rating:** 5
**Confidence:** 4

**Summary:**

This paper proposes Action Temporal Coherence Learning (AcTOL), a vision-language pretraining method for embodied agents. AcTOL addresses limitations in prior goal-reaching alignment approaches (e.g., misalignment due to irrelevant video frames) by enforcing temporal ordering via a novel Vision-Language Ordering loss and continuity via a Brownian Bridge constraint.
Theoretical guarantees for ordering, continuity, and language robustness are provided. Experiments on real and simulated robots show significant improvements in language-conditioned imitation learning, especially with limited demonstrations. AcTOL also enables accurate language-conditioned visual rewards and exhibits strong robustness to linguistic perturbations.

**Questions:**

- How does AcTOL handle long-horizon tasks with multiple subgoals (e.g., "make coffee")? Does the Brownian Bridge scale to such sequences?

- Why was the Brownian Bridge loss weight fixed to 0.1? Was sensitivity tested for different tasks?

- For real-world deployment: How does AcTOL address visual domain gaps (e.g., human hands vs. robot grippers) when pretrained on human videos?

**Ethical Concerns:**

["NO or VERY MINOR ethics concerns only"]

**Final Justification:**

The clarifications have satisfactorily addressed my concerns, and I have no further questions. I will raise my rating to 5.

**Limitations:**

yes

**Quality:**

3

**Strengths And Weaknesses:**

Strengths
- The VLO loss and Brownian Bridge constraint offer a principled solution to noisy action boundaries, avoiding rigid goal-reaching assumptions.

- Provable guarantees for ordering (Theorem 1), continuity (Theorem 2), and language-perturbation robustness (Theorem 3) strengthen the methodology.

-  The proposed method significantly outperforms baselines (LIV, R3M, DecisionNCE) in simulation and real-robot tasks; excels in low-data regimes; generates dense instruction-aligned visual rewards; and maintains robustness under diverse linguistic instructions.

Weaknesses
- Assumption of linear temporal progression fundamentally conflicts with cyclic/repetitive actions (e.g., dishwashing "scrub - rinse - scrub" cycles), limiting applicability to such tasks, with no experimental validation provided.

- Spatial perception remains a critical bottleneck, evidenced by the 50% real-robot success rate for geometrically complex tasks (cup-picking vs. ~80% for drawers; Table 2), as CLIP-based frozen encoders fail to capture fine-grained affordances like handle orientation.

---

> ### Author Rebuttal · Authors · 2025-07-29
>
> Thank you for your thoughtful feedback! Below, we provide point-by-point responses to each of your questions.
> > **W1. Limitation with cyclic or repetitive actions.**
>
> Our method relies on the assumption that actions progress in a temporally ordered manner. **As we have mentioned in Limitations (line 736)**, repetitive, such as dishwashing (e.g., scrub, rinse, scrub), may partially violate this assumption and could pose challenges for our current formulation.
>
> However, most real-world tasks, particularly in household robotics, tend to follow a primarily ordered structure. Our approach is therefore well-suited for such scenarios, as demonstrated by the empirical results. We appreciate the suggestion and consider handling repetitive or cyclic behaviors an important direction for future work.
>
> > **W2. Spatial perception and fine-grained affordance issues.**
>
> We agree that spatial perception plays a significant role in geometrically complex tasks. While CLIP-based encoders offer strong semantic priors, they may lack the fine-grained spatial precision needed for tasks like cup-picking.
>
> However, our work primarily focuses on improving the temporal ordering and continuity of visual-language representation, **rather than object- or pixel-level spatial understanding**. That said, enhancing spatial perception is a promising direction to complement our approach. For instance, some **feature-level super-resolution techniques** like Featup [1] could be explored to improve the spatial granularity of CLIP embeddings, combining our method to achieve temporal and spatial optimization.
>
> [1] FeatUp: A Model-Agnostic Framework for Features at Any Resolution, ICLR'24
>
>
> > **Q1. Scalability to long-horizon, multi-subgoal tasks.**
>
> AcTOL is not specifically designed for long-horizon tasks with multiple subgoals, but it can be naturally extended to such settings by leveraging a Large Language Model (**LLM**) as **a high-level planner**, like [2]. The LLM can decompose a complex instruction (e.g., "make coffee") into a sequence of shorter sub-tasks (e.g., "pick up kettle" → "fill water" → "boil" ...), each of which can then be independently handled by AcTOL.
>
> Regarding the scalability of the Brownian Bridge loss, Figure 4 demonstrates that AcTOL **maintains smooth transitions even between distinct sub-tasks**. This continuity property is expected to generalize to longer sequences of consecutively executed tasks. We will include additional visualizations in the revised version to further support this point.
>
> [2] Language Models as Zero-Shot Planners: Extracting Actionable Knowledge for Embodied Agents, ICML'22
>
>
> > **Q2. Hyperparameter sensitivity analysis ($\lambda$).**
>
> Due to space constraints, we included the sensitivity analysis of the hyperparameter $\lambda$ in **Appendix A (line 570)**. As shown in the results, our strategy is verified agnostic to the choices of $\lambda$ varying from ${0.01}$ to ${1}$. We chose $0.1$ to ensure a reasonable scaling between the ordering and continuity losses.
>
> > **Q3. Visual domain gaps between human videos and real-world deployment.**
>
> Thank you for raising this important point. There is indeed a visual domain gap between human-centric videos and third-person robot observations. AcTOL can addresses this challenge in belowing ways.
> * AcTOL learns a **general** inductive bias of temporal ordering and continuity from vision-language pretraining. This bias focuses on the progression of visual states over time, **rather than on specific visual details**. As a result, the learned features can generalize across different embodiments. For example, although a robot gripper may look different from a human hand, both follow similar patterns of interaction when performing tasks like opening a drawer or picking up an object. This generalization ability is demonstrated in Figures 4 and 9, where AcTOL successfully produces **meaningful reward signals from real-world robot videos, even when trained on human videos**.
> * During downstream policy learning such as behavior cloning, we find only **fine-tune the model using a small number of robot demonstrations** can largely mitigate the domain gap. To prove this, we first take 25 in-domain demonstrations (5 per task) in Franka Kitchen to **fine-tune** the pre-trained encoders using the AcTOL objective. Then, as before, we freeze the fine-tuned encoders and train policies on top using behavior cloning. We report the comparisons when using 15 demos for policy training. As shown in the table below, the success rate improvement also demonstrate that the learned temporal inductive bias can be **effectively adapted to the robot domain with limited supervision**.
>
>
> | Franka Kitchen | Frozen SR| Finetune SR|
> | -------- | -------- | -------- |
> | AcTOL     |   61.8   |  **86.4**  |

---

> > ### Comment · Reviewer_Mtcd · 2025-08-02
> >
> > Thank you for your detailed response. The clarifications have satisfactorily addressed my concerns, and I have no further questions. I will raise my rating to 5.

---

### Official Review · Reviewer_7Q3R · 2025-07-02

**Clarity:** 3
**Significance:** 4
**Originality:** 4
**Rating:** 5
**Confidence:** 2

**Summary:**

This paper introduces **Action Temporal Coherence Learning (AcTOL)**, a vision-language pre-training framework designed for embodied agents. AcTOL tackles the traditional gap in rigid goal-based constraints from traditional video pretraining.

1. **Vision-Language Ordering (VLO) loss**: contrasts pairs of frames based on their temporal distance, ensuring semantic alignment respects natural ordering.
2. **Brownian bridge continuity constraint**: models local frame intervals as a Brownian bridge, enforcing smooth transitions in the learned visual feature space.

The authors provide theoretical guarantees for both ordering and continuity, and demonstrate via imitation-learning on simulated (Franka Kitchen, MetaWorld) and real (Unitree D1) robotic tasks that AcTOL outperforms existing methods (e.g., LIV, DecisionNCE, R3M), especially in low-data regimes

**Questions:**

I do wonder, tho, does the algorithm assume that the video is temporally progressive? Like an action is executed from start to finish, instead o,f say, reversed or temporally inconsistent? If such a video is feeded to the algorithm, what would happen?

**Ethical Concerns:**

["NO or VERY MINOR ethics concerns only"]

**Final Justification:**

I will maintain my original judgment after the authors provide answers to my questions

**Limitations:**

Yes

**Quality:**

4

**Strengths And Weaknesses:**

## Strength

- Interesting formulation of the problem, aiming at a meaningful gap in the community
- Rigorous theoretical analysis
- Exhaustive empirical study

## Weakness
- No obvious weakness.

I will admit that due to the overwhelming review responsibility this year, I have not had enough time to check all the math in the paper. I will continue to check them after the review deadline.

---

> ### Author Rebuttal · Authors · 2025-07-29
>
> Thank you for your thoughtful feedback!
> > **Q1. Assumption of temporal progression in videos.**
>
>  Yes, our method assumes that input videos are temporally progressive. If temporally inconsistent or reversed videos are introduced during pretraining, they could potentially degrade the model’s learning, since the visual transitions would no longer reflect meaningful action progressions. However, we believe such cases are rare in naturally collected human demonstration videos.
>
> At evaluation time, our trained model is capable of identifying inconsistencies between the video content and the given instruction. For example, when prompted with an instruction like "open the drawer", if a **reversed video** (i.e., a drawer being closed) is provided, the model tends to assign **a decreasing reward**, as shown in Figure 4. If a video contains **unrelated content** (e.g., washing dishes) or **shuffled frames**, AcTOL produces **fluctuating reward curves** that reflect poor alignment with the intended task. We appreciate this suggestion and will include additional examples of such cases in the revised version of the paper to further illustrate the model's behavior.

---

> > ### Comment · Reviewer_7Q3R · 2025-08-04
> >
> > Thank you for your response. I will maintain my rating of 5.

---

### Official Review · Reviewer_h3Cm · 2025-07-03

**Clarity:** 3
**Significance:** 2
**Originality:** 3
**Rating:** 4
**Confidence:** 3

**Summary:**

This paper proposes AcTOL, a vision-language pretraining framework that leverages temporal ordering (via a Vision-Language Ordering loss) and temporal continuity (via a Brownian bridge constraint) to learn more robust and transferable visual-language representations from egocentric human action videos. The motivation is to reduce the dependence on expensive, fully-annotated robot trajectories for imitation learning. AcTOL is validated through language-conditioned behavior cloning on both real and simulated robot tasks, along with reward learning experiments and language perturbation tests. The results demonstrate improved data efficiency and robustness compared to prior methods.

**Questions:**

1. How do you mitigate the domain gap between egocentric pretraining data and third-person robot observations?
2. Does the method remain robust if significant occlusions, distractors, or dynamic environment changes are present?
3. The hyperparameter λ controlling the trade-off between the ordering and continuity terms is fixed at 0.1, but no sensitivity analysis is presented. Did the authors examine how varying λ impacts training stability and performance?
4. The Brownian bridge assumes smooth transitions across video frames, yet real-world tasks often exhibit clear multi-stage or discrete transitions (e.g., “grasp → lift → place”). Could the bridge oversmooth critical semantic boundaries? Have the authors considered segment-wise or adaptive bridge constraints to better capture such multi-phase actions?
5. The VLO loss defines negative samples purely by temporal distance. In videos containing repeated or cyclic actions (e.g., repeatedly opening/closing a drawer), temporally distant frames might share nearly identical semantics. Could the authors clarify how reliable temporal distance is in such repetitive scenarios, and whether any alternative negative sampling strategies were explored?

**Ethical Concerns:**

["NO or VERY MINOR ethics concerns only"]

**Final Justification:**

The extra analyses and experiments have effectively addressed most of the concerns I brought up, so I have decided to raise the score.

**Limitations:**

Overall, the authors have not sufficiently addressed the broader limitations of their method or the potential societal risks, such as domain transfer challenges and biases in human demonstration data, which warrant more thorough discussion.

**Paper Formatting Concerns:**

I did not identify any major formatting problems in the paper. However, I encourage the authors to carefully proofread for minor typos or spacing inconsistencies before final submission.

**Quality:**

3

**Strengths And Weaknesses:**

**Strengths**
1. Experiments span real and simulated robots, with reasonable baselines and ablation comparisons.
2. The mathematical analysis provides insights into the expected temporal consistency properties of the learned features.

**Weakness**
1. Real-robot experiments are limited to three fairly simple tasks in controlled conditions, without diverse backgrounds, occlusion, or dynamic disturbances.
2. There is a significant domain gap between egocentric pretraining data and third-person real-robot evaluation, which is insufficiently analyzed.
3. The language perturbation coverage is narrow and focuses on short phrases, lacking complex compositional instructions.

---

> ### Author Rebuttal · Authors · 2025-07-29
>
> Thank you for your thoughtful feedback! Below, we provide point-by-point responses to each of your questions.
>
> > **W1/Q2. Robustness under visual shifts.**
>
> We appreciate the reviewer’s concern regarding the limited diversity in our real-world experiments. Due to limited time, we were unable to collect additional data and complete a full evaluation of visual shift robustness in the real-world setting. To partially address this concern, we have conducted **additional experiments in the Franka Kitchen environment by introducing visual distribution shifts** following the setup in reference [1]. In particular, we evaluate AcTOL and the strongest baseline method, DecisionNCE, under various visual changes that are not present in the training data. These changes include:
> * **Object distractors** of increasing difficulty: easy, medium and hard level, corresponding to scenes containing 1, 3, and 9 distracting objects from YCB object set, respectively.
> * **Texture variations** in the background: marble hinge texture and metal slide texture.
>
> We use 15 demonstrations per task for policy training. The success rates averaged over 5 tasks under each setting are presented in the following table:
> |Method|D(easy)|D(medium)|D(hard)|T(marble hinge)|T(metal slide)|No shift|
> |-|-|-|-|-|-|-|
> |DecisionNCE|27.2|25.6|4.8 |0 |8.8| 43.2|
> |AcTOL|**43.2**|**32.8**|**9.2**|**4.4**|**38.4**|61.8|
>
> While performance drops under visual shifts, which is expected, AcTOL continues to **outperform DecisionNCE in all available test conditions**. This suggests that the learned representation maintains useful generalization ability even without any specific adaptation for visual domain shift.
>
> For future work, we plan to improve visual robustness by incorporating stronger image backbones such as Vision Transformers, applying more extensive data augmentation during pretraining, and exploring domain randomization techniques to enhance performance under visual out-of-distribution conditions.
>
> [1] WHAT MAKES PRE-TRAINED VISUAL REPRESENTATIONS SUCCESSFUL FOR ROBUST MANIPULATION? Arxiv'23
>
>
> > **W2/Q1. Domain gap between egocentric pretraining and third-person deployment.**
>
> Thank you for raising this important point. There is indeed a visual domain gap between human-centric videos and third-person robot observations. AcTOL can addresses this challenge in belowing ways.
> * AcTOL learns a **general** inductive bias of temporal ordering and continuity from vision-language pretraining. This bias focuses on the progression of visual states over time, **rather than on specific visual details**. As a result, the learned features can generalize across different embodiments. For example, although a robot gripper may look different from a human hand, both follow similar patterns of interaction when performing tasks like opening a drawer or picking up an object. This generalization ability is demonstrated in Figures 4 and 9, where AcTOL successfully **produces meaningful reward signals from real-world robot videos, even when trained on human videos**.
> * During downstream policy learning such as behavior cloning, we find only **fine-tune the model using a small number of robot demonstrations** can largely mitigate the domain gap. To prove this, we first take 25 in-domain demonstrations (5 per task) in Franka Kitchen to **fine-tune** the pre-trained encoders using the AcTOL objective. Then, as before, we freeze the fine-tuned encoders and train policy networks on top using behavior cloning. We report the comparisons when using 15 demos for policy training. As shown in the table below, the success rate improvement also demonstrate that the learned temporal inductive bias can be **effectively adapted to the robot domain with limited supervision**.
>
> | Franka Kitchen | Frozen SR| Finetune SR|
> | -------- | -------- | -------- |
> | AcTOL     |   61.8   |  **86.4**  |
>
> > **W3. Narrow language perturbation coverage.**
>
>
> We appreciate the reviewer’s concern regarding the evaluation of language diversity. As detailed in **Appendix B.4 (line 609)**, we have included an experiment specifically designed to assess this aspect. In this setting, we provide four language variations per task, including those generated by GPT-4o. These instructions are approximately **15** words long and feature **diverse verbs, richer noun phrases, and extended contexts**. For example:
>
> > *"Mind pushing open the right cupboard cabinet door? I need to grab the cups inside."*
>
> Despite the increased linguistic complexity, AcTOL experiences only a modest **3% drop** in success rate (see Appendix B.4) while still outperforming baseline methods. This experiment is intended to emulate the variability of natural household instructions, and we believe it offers a meaningful measure of robustness under diverse language conditions.
>
>
> > **Q3. Hyperparameter sensitivity analysis ($\lambda$).**
>
> Due to space constraints, we included the sensitivity analysis of the hyperparameter $\lambda$ in **Appendix A (line 570)**. As shown in the results, our strategy is verified agnostic to the choices of $\lambda$ varying from ${0.01}$ to ${1}$. We chose $0.1$ to ensure a reasonable scaling between the ordering and continuity losses.
>
> > **Q4. Brownian Bridge might oversmooth critical semantic boundaries.**
>
> While real-world tasks often consist of multiple discrete semantic stages (e.g., “grasp → lift → place”), these stages typically unfold in **a continuous and temporally ordered manner**. As a result, visual features near stage boundaries usually exhibit gradual transitions rather than abrupt changes. As such, visual features near stage boundaries tend to exhibit gradual transitions rather than abrupt discontinuities.
>
> To capture this structure, our approach employs two complementary components: **VLO** captures **long-term visual-semantic ordering** and helps **recognize discrete semantic boundaries for multi-stage actions**, while Brownian bridge act at a **short-term frame level**, promoting **smooth local transitions in visual representations**. This division of roles ensures that high-level semantic stages are preserved, while still allowing dense visual reward learning at a finer timescale.
>
> As illustrated in Figure 4, AcTOL successfully distinguishes between different action phases while maintaining temporal continuity. This continuity is beneficial for producing **dense reward signals that vary smoothly with task progress**, leading to more effective learning in complex scenarios.
>
> > **Q5. Limitations of VLO loss with repetitive actions.**
>
> Thanks. The datasets we use for pretraining, such as EPIC-KITCHEN, consist of clips that focus on **a single activity segment**, and therefore do not contain cyclic or repetitive actions like repeatedly opening and closing a drawer.
>
> However, when using noisy web videos that may include cyclic behaviors, we acknowledge that temporally distant frames can occasionally share similar semantics, potentially affecting the training performance as **we have mentioned in Limitations (line 736)**.
>
> In practice, this issue is somewhat alleviated since **chance of sampling semantically overlapping frames as negatives is relatively low**.
>
> To further reduce this risk, we propose a few simple but effective strategies:
>
> * **Similarity filtering**: discard candidate negatives that are too similar to the positive.
> * **Action-boundary segmentation**: segment actions before training to avoid sampling within the same repeated behavior.
>
> For future work, we plan to explore how noisy videos with cyclic or repetitive actions can be better incorporated into AcTOL pretraining.

---

> > ### Comment · Reviewer_h3Cm · 2025-08-06
> >
> > Thank you for your detailed responses to the review comments. The additional analyses have effectively addressed most of the concerns raised, and the supplementary experiments provide stronger empirical support for the claims regarding AcTOL’s robustness and generalization. In light of these improvements, I have decided to raise my score.  I encourage the authors to incorporate the clarifications and new results presented in the rebuttal into the final version of the paper to further strengthen its contribution.

---

### Note · Authors · 2025-08-12

We sincerely thank the program committee members and all reviewers for their time and constructive feedback! The insightful comments and discussions have been valuable in improving our work. We will incorporate the suggested revisions and clarifications in the final version to better present our contributions.

---

### Decision · Program_Chairs · 2025-09-17

**Decision:**

Accept (poster)

**Comment:**

**Summarization** This paper proposes Action Temporal Coherence Learning (AcTOL), which treats videos as continuous trajectories. AcTOL contrasts semantic differences between frames to capture temporal order and applies a Brownian bridge constraint to ensure smooth transitions. Experiments on simulated and real robots show that AcTOL-pretrained features enhance manipulation tasks and remain robust to diverse linguistic instructions, paving the way toward generalized embodied agents.

**Strengths**
1. The mathematical analysis provides deep insights into the temporal consistency properties.
2. The VLO loss and Brownian Bridge constraint offer a principled solution to noisy action boundaries
3. Very good performance, showing the promising manner for affordable pre-training for robots.

**Weakness**

The main concerns are about long-horizon tasks and the visual domain gap. But the authors have well addressed these concerns in rebuttal, such as both the proposed continuity constraint and the combination with existing task planners can mitigate the long-horizon challenges. Regarding the domain gap, the proposed method can learn a general representation, and experimental results show very good generalization with a few data finetuning.

**Justification for Decision**

The proposed method is technically novel and demonstrated to be very effective, which makes a significant contribution in the direction of video-based robot pre-training. Furthermore, the theoretical analysis provides deep insights. All the raised concerns have been well addressed by the authors. Therefore, I made the decision to accept. The authors should incorporate the rebuttal into their final version.